# FaceMoE: Mixture of Experts for Low-Resolution Face Recognition

## Abstract

Low-resolution face recognition (LR-FR) remains a challenging task due to poor feature extraction and aggregation, as probe images often contain limited identity information resulting from extreme degradations such as blur, occlusion, and low contrast. Additionally, the domain gap between high-resolution (HR) gallery images and low-resolution (LR) probe images poses a significant challenge. A single feature encoder struggles to generalize effectively across both domains when fine-tuned on an LR dataset, and this issue is further magnified by catastrophic forgetting. To address these challenges, we propose **FaceMoE**, a novel transformer-based architecture enhanced with a Mixture of Experts (MoE) design. Specifically, we introduce multiple specialized feed-forward network (FFN) experts and incorporate a top-$k$ router, which dynamically assigns tokens to appropriate experts. This design promotes specialization across experts for different semantic regions of the face, which enables FaceMoE to perform *resolution-aware feature extraction*. Moreover, the top-$k$ router facilitates sparse expert activation, enabling the model to preserve pretrained knowledge when finetuned on a LR dataset, while increasing model capacity without proportional computational overhead. FaceMoE is trained with a combined face recognition loss, router $z$-loss, and load balancing loss to ensure expert specialization and stable training. To the best of our knowledge, this is the first work leveraging MoE for LR-FR. Extensive experiments across eleven datasets, spanning HR, mixed-quality, and LR benchmarks, demonstrate that Face-MoE significantly outperforms state-of-the-art methods, excelling in low-resolution face recognition. Code and models will be made public.

## 1 Introduction

Face recognition is one of the foundational tasks in computer vision and biometrics, involving the recognition and verification of individuals from images or videos. It plays a vital role in real-world applications such as authentication Roy et al. (2025), banking Vishnuvardhan & Ravi (2021), and border control Hidayat et al. (2024). Recently, there has been a growing focus on low-resolution face recognition (LR-FR) Cheng et al. (2019); Chai et al. (2023); Jawade et al. (2024a), due to its widespread applicability in surveillance Kalka et al. (2018). However, this task is particularly challenging because the input images or videos are often of surveillance quality and severely degraded by factors such as atmospheric turbulence, occlusion, overexposure, and motion blur. These degradations significantly reduce the discriminative features necessary for reliable identification, making conventional recognition techniques less effective. Additionally, variations in pose, illumination, and expression become more pronounced and harder to manage in low-resolution settings, often resulting in poor generalization and reduced performance. Therefore, LR-FR remains a challenging yet crucial problem to address.

To improve the effectiveness of low-resolution face-recognition, it is essential to address several key challenges: ***Challenge 1 - Effective face feature aggregation***: Probe videos in low-resolution datasets often suffer from significant degradation, which makes face feature aggregation particularly difficult. Since only a limited subset of frames typically contains discriminative identity information, effective feature extraction, followed by aggregation is crucial to build robust face templates. ***Challenge 2 - HR gallery and LR probe domain difference***: In LR-FR, gallery images are typically high-resolution (HR), while probe images are low-resolution (LR) and come from distinct domains, as validated in Figures 1(a) and 1(b). Models tend to rely on different semantic regions depending on the input

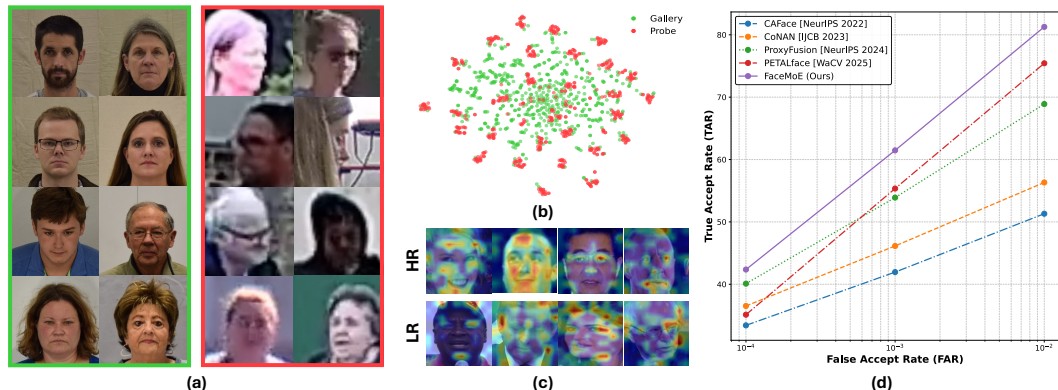

Figure 1: (a) BRIAR gallery and probe. (b) Domain difference between gallery and probe. (c) Activation maps corresponding to LR and HR images. (d) SOTA results on BRIAR Protocol 3.1.

resolution to achieve accurate recognition. For HR images, they focus on skin texture, landmarks regions and other fine details that provide sufficient identity information. In contrast, for LR inputs, the face region can be severely degraded to extract any identity information. In such cases, the focus shifts towards broader shapes and coarse facial structures. These resolution-dependent patterns are clearly illustrated by the activation maps in Figure 1(c). This gallery and probe domain gap poses a significant challenge for effective feature extraction. ***Challenge 3 - Catastrophic forgetting when adapting to LR dataset:*** LR-FR models are generally trained in two stages: large-scale pretraining on HR datasets, followed by finetuning on the target LR domain. The second-stage adaptation process makes the model prone to *catastrophic forgetting*, due to unstable gradient updates in the initial epochs of finetuning Li & Hoiem (2017), caused by the significant domain difference between HR and LR datasets. As a result, the model not only loses its pretrained performance but also fails to effectively adapt to the low-resolution data. We validate this effect empirically and show the resulting performance drop in Figure 3 (Finetuning (CosFace)) and Figure 4 (Finetuned CosFace).

Existing works aimed at improving low-resolution face recognition, such as CAFace Kim et al. (2022b), CoNAN Jawade et al. (2024a), and ProxyFusion Jawade et al. (2024b), focus on addressing *Challenge 1* by selecting relevant frames for fusion after the feature extraction. Specifically, CAFace Kim et al. (2022b) utilizes an intermediate style map; CoNAN Jawade et al. (2024a) learns a context vector conditioned on distributional information to weigh features based on their estimated informativeness; and ProxyFusion Jawade et al. (2024b) employs learnable queries to identify the most relevant frames. However, the effectiveness of these methods is constrained by the quality of the trained feature encoders, which ultimately limits their overall performance. In contrast, PETAL*face* Narayan et al. (2025) introduces quality-adaptive dual low-rank modules aimed at developing a more generalized feature encoder across both high-resolution and low-resolution domains, thereby catering to all the key challenges in low-resolution face recognition. Nevertheless, its performance on the low-resolution domain remains subpar compared to the other SOTA methods.

To address the aforementioned challenges, we propose **FaceMoE**, a novel framework designed to tackle the core issues in LR-FR. We introduce an architectural modification to the transformer block by incorporating a mixture of feed-forward network (FFN) experts in place of the standard single FFN. Existing transformer-based face recognition encoders typically employ a single FFN following the self-attention operation. However, we argue that a single FFN is insufficient for the complex task of low-resolution face recognition, as it struggles to effectively handle both the HR gallery and LR probe domains. Moreover, it lacks the *resolution-aware feature extraction* necessary for robust identity representation. Our modified transformer block addresses these limitations by using multiple FFN experts and a top-$k$ router that directs each input patch to a subset of $k$ out of $n$ experts based on the input resolution. This design enables different FFN experts to specialize in distinct facial regions, with the top-$k$ router dynamically assigning the subset of experts based on resolution, thereby achieving *resolution-aware feature extraction*. This enables the model to extract strong identity representations by routing input tokens from regions that retain identity cues in degraded images to specialized experts tailored to those regions. This improves feature extraction from LR probes and enhances overall face feature aggregation. Furthermore, the presence of multiple FFN experts facilitates effective adaptation to LR datasets with a minimal drop in pretrained performance.

This is achieved through the modular and sparsely activated nature of MoE, which restricts weight updates to only a subset of experts during fine-tuning, thereby reducing *catastrophic forgetting* Fedus et al. (2022). The modular design allows experts to function as semi-independent blocks; during fine-tuning, this structure induces *selective drift* Rypeść et al. (2024), with some experts adapting to LR data while others retain their pretrained knowledge. This retained knowledge enables the model to perform effective feature extraction for both the HR gallery and LR probe domains, further enhanced by its resolution-aware feature extraction capabilities. FaceMoE is trained using a combination of router $z$-loss and load-balancing loss, which promotes both expert specialization and balanced utilization, thereby preventing training collapse. The top-$k$ routing ensures sparse expert utilization, with a increase in model capacity without a proportional rise in computational cost achieving $2.17\times$ more capacity with only $1.66\times$ more FLOPs.

To summarize our contributions are as follows:

1. We propose **FaceMoE**, a modified transformer encoder with sparsely activated FFN experts. It enables efficient adaptation to low-resolution datasets while minimizing *catastrophic forgetting*, effectively addressing the domain gap between gallery and probe images.
2. We introduce a top-$k$ router that assigns each input token to a subset of FFN experts, each specializing in distinct semantic facial regions. This enables *resolution-aware feature extraction*. The router directs tokens containing discriminative identity cues to the most relevant experts, thereby enhancing feature representation and improve LR-FR performance.
3. We demonstrate the effectiveness of FaceMoE by outperforming state-of-the-art models on low-resolution face recognition (see Figure 1(c)). We showcase its capabilities through evaluations on eleven datasets, covering HR, mixed-quality, and LR scenarios.

## 2 RELATED WORK

**Low Resolution Face-Recognition.** Face recognition research has largely focused on developing variants of margin-based loss functions Deng et al. (2019); Wang et al. (2018); Liu et al. (2017); Wen et al. (2016); Kim et al. (2022a); Huang et al. (2020b) to improve the performance on high-resolution benchmarks Cheng et al. (2019); Kalka et al. (2018); Cornett et al. (2023). In contrast, much less attention has been given to low-resolution unconstrained face recognition (LR-FR) datasets Cheng et al. (2019); Kalka et al. (2018); Cornett et al. (2023), which contain heavily degraded face images that are unidentifiable by humans. Efforts to improve LR-FR can be broadly categorized into four areas based on their focus: data, training methodology, feature fusion, and architectural design. Early works Singh et al. (2018); Yue et al. (2016) used super-resolution (SR) models to restore images prior to recognition, but later works Li et al. (2019); Zhang et al. (2018); Jiang et al. (2018) suggest that this approach can cause identity hallucination. Many studies Hsu et al. (2019); Yin et al. (2020); Yu et al. (2018); Singh et al. (2021) relate recognition to visual quality. However, this is infeasible as it requires paired HR and LR images of the same subject, which are mostly unavailable in LR datasets. Low & Teoh (2022); Low et al. (2021) introduce augmentations to mitigate the performance gap between HR and LR samples. In terms of training methods, some works Massoli et al. (2020); Zhu et al. (2019) use knowledge distillation to transfer information from the HR domain to the LR domain. For instance, Ge et al. (2018; 2020) adopt a teacher-student framework, while Huang et al. (2020a) proposes a distribution distillation loss. Additionally, Chai et al. (2023) focuses on optimizing the embedding space to boost performance. In the area of feature fusion, CAFace Kim et al. (2022b) proposes a two-stage approach that leverages style information. CoNAN Jawade et al. (2024a) learns a context vector conditioned on the distribution and weighs features based on their estimated informativeness. ProxyFusion Jawade et al. (2024b) employs learnable queries to select a sparse set of expert networks for feature aggregation. Recent architecture-based methods include PETALface Narayan et al. (2025), which introduces two image quality-adaptive LoRA modules. Our work, FaceMoE, also falls within the architecture category. We introduce multiple FFN experts, each specialized in different face regions for enhanced feature encoding. This design achieves state-of-the-art performance on multiple low-resolution face recognition benchmarks.

**Mixture of Experts.** Mixture of Experts (MoE) architectures have emerged as a powerful approach to scale model capacity efficiently by activating only a subset of specialized experts per input. Shazeer et al. (2017) introduced sparsely-gated MoEs, demonstrating their effectiveness in large language models. Lepikhin et al. (2020) employed conditional computation and automatic sharding

to scale transformer-based models to the trillion-parameter range through efficient model and data parallelism. Several works have adopted the MoE design for vision applications such as image classification Riquelme et al. (2021); Han et al. (2024); Zhang et al. (2024); Puigcerver et al. (2023), object detection Oksuz et al. (2023); Jain et al. (2023), semantic segmentation Wang et al. (2020); Rossi et al. (2025), and image generation Xue et al. (2023); Park et al. (2018); Jiang et al. (2022). Building on these advances, recent efforts have also explored MoE architectures for face-related applications. MoE-FFD Kong et al. (2024) proposes a parameter-efficient ViT-based approach for face forgery detection by integrating MoE modules with LoRA and adapter layers. Zhou et al. (2022) presents a MoE-injected architecture with a dynamic expert aggregation network for generalizable face anti-spoofing. In our work, we aim to use an MoE-enhanced transformer architecture to boost the performance of LR-FR.

## 3 METHOD

In this work, we aim to enhance the generalization capability of face recognition models, with a particular focus on improving LR-FR performance. We first introduce preliminary concepts regarding Mixture of Experts. We then propose FaceMoE, an MoE-enhanced transformer that facilitates robust feature extraction across both HR and LR domains, while mitigating catastrophic forgetting when fine-tuned on LR datasets. Finally, we outline our training framework for stable convergence.

### 3.1 PRELIMINARIES: MIXTURE OF EXPERTS

The MoE framework Jacobs et al. (1991); Shazeer et al. (2017) is a modular neural architecture that leverages multiple specialized sub-models (experts) to model complex data distributions. Formally, let $x \in \mathbb{R}^d$ be an input vector. The MoE model consists of $N$ experts $\{f_i(x; \theta_i)\}_{i=1}^N$, where each $f_i : \mathbb{R}^d \to \mathbb{R}^m$ is parameterized by $\theta_i$, and a gating network $G(x; \phi) = [w_1(x), \dots, w_N(x)]$, parameterized by $\phi$, which outputs a probability distribution over the experts such that $\sum_{i=1}^N w_i(x) = 1$. The gating weights are commonly obtained using a softmax, defined as $w_i(x) = \frac{\exp(g_i(x))}{\sum_{j=1}^N \exp(g_j(x))}$, where $g_i(x)$ denotes the score of the $i$-th expert. The final output of the MoE is a convex combination of the expert outputs, given by

$$y = \sum_{i=1}^N w_i(x) f_i(x; \theta_i).$$

The training objective minimizes a loss function $\mathcal{L} = \frac{1}{K} \sum_{k=1}^K \ell\left(y^{(k)}, \sum_{i=1}^N w_i(x^{(k)}) f_i(x^{(k)}; \theta_i)\right)$, where $\ell(\cdot, \cdot)$ is a task-specific loss (such as mean squared error or cross-entropy). Sparse MoE variants Shazeer et al. (2017) further improve computational efficiency by restricting active experts to a subset $S \subset \{1, \dots, N\}$, yielding $y = \sum_{i \in S} w_i(x) f_i(x; \theta_i)$. In this work, we propose Face-MoE, which adopts the MoE paradigm within a transformer-based FR model to enable dynamic routing and specialization across experts, thereby enhancing feature extraction and improving LR-FR performance.

### 3.2 FACEMOE

To address the challenge of feature extraction in LR-FR, we introduce FaceMoE, a novel transformer architecture enhanced with an MoE mechanism. The primary motivation behind integrating MoE within the transformer blocks is to encourage dynamic specialization of sub-networks (experts) to different patterns present in facial data. FaceMoE inserts the experts into the feed-forward (MLP) layers. We select linear projections as experts due to their proven capacity to introduce additional non-linearity when composed with transformer self-attention, enhancing the model's ability to capture complex patterns in data Vaswani et al. (2017). The linear layer experts serve to extract complementary information from the attended tokens generated by the multi-head self-attention operation. This design choice balances expressiveness and computational efficiency, as the MLP layers constitute a significant portion of transformer model capacity. This modular approach allows the model to adapt better to low-resolution face images, while preserving pretrained knowledge.

**Mixture of Experts MLP Layer:**

In FaceMoE, the MoE is incorporated inside the MLP layers of the transformer block. Let $x \in \mathbb{R}^{T \times d}$ represent a sequence of $T$ tokens, each of dimensionality $d$, output by the self-attention block. The expert layer comprises $N$ expert MLPs, $\{f_i(x; \theta_i)\}_{i=1}^{N}$, each parameterized by weights $\theta_i$. The experts operate independently but in parallel to process the input tokens. An individual expert is a two-layer fully connected network with weights $\{W_{i,1}, W_{i,2}\}$ and biases $\{b_{i,1}, b_{i,2}\}$, defined as:

$$f_i(x_t) = W_{i,2} \cdot \sigma(W_{i,1} x_t + b_{i,1}) + b_{i,2}, \quad \forall t \in \{1, \dots, T\},$$

where $\sigma(\cdot)$ is an activation function, in this case GELU Hendrycks & Gimpel (2016), $W_{i,1} \in \mathbb{R}^{d \times h}$, $W_{i,2} \in \mathbb{R}^{h \times d}$, $b_{i,1} \in \mathbb{R}^h$, and $b_{i,2} \in \mathbb{R}^d$, with $h$ being the hidden dimension. This formulation enables each expert to non-linearly transform and project each token representation.

**Top-$k$ Router:**

The top-$k$ router is a core component of FaceMoE, responsible for dynamically assigning input tokens to a subset of experts. Given token embeddings $x \in \mathbb{R}^{T \times d}$, the router computes expert selection logits for each token $x_t$ using a linear projection: $z_t = x_t W_r$, where $W_r \in \mathbb{R}^{d \times N}$ are learnable routing weights, and $z_t \in \mathbb{R}^N$ contains the routing scores for the $N$ experts. For each token $t$, the router selects the indices of the top-$k$ experts with the highest activations: $(i_1, i_2, \dots, i_k) = \text{TopK}(z_t)$, where $i_j \in \{1, \dots, N\}$. The logits of the selected experts are normalized by a softmax over the top-$k$ values to produce the routing probabilities: $w_{i_j}(x_t) = \frac{\exp(z_{t,i_j})}{\sum_{j=1}^{k} \exp(z_{t,i_j})}$. The final output of the MoE layer for token $x_t$ is a convex combination of the outputs of the selected experts: $y_t = \sum_{j=1}^{k} w_{i_j}(x_t) f_{i_j}(x_t)$. This sparse routing strategy leads to significant computational savings, as only $k < N$ experts are active per token. Importantly, it enables efficient adaptation to low-resolution datasets. In our experiments, we empirically found that setting $N = 3$ and $k = 2$ yielded the best trade-off between model performance and efficiency. Under this configuration, we observed that the router exhibits conditional routing behavior, where each expert is implicitly specialized for certain semantic regions of the face, as shown in Figure 2. This behavior can be expressed by the conditional routing probability:

$$\mathbb{P}(i_j \mid R_t = r) > \mathbb{P}(i_j \mid R_t \neq r), \quad \forall r \in \{\text{high-freq}, \text{low-freq}, \text{landmarks}\},$$

where $R_t$ denotes the semantic or frequency region of token $x_t$. Specifically, tokens corresponding to high-frequency regions (e.g., edges, contours, hair textures, background) are primarily routed to one expert; tokens from low-frequency smooth regions (e.g., cheeks, forehead) are directed to a second expert; and tokens corresponding to landmark regions (e.g., eyes, nose) are routed to the third expert.

### 3.3 TRAINING FRAMEWORK

To train FaceMoE, we optimize a composite objective combining a primary face recognition loss with auxiliary regularization terms designed to stabilize the MoE routing process. The primary loss is based on the well-established *CosFace* margin-based softmax loss Wang et al. (2018) denoted as $\mathcal{L}_{\text{face}}$, which encourages inter-class separability and intra-class compactness in the learned embedding space. In addition, we introduce two auxiliary losses applied to the router network:

**1. Router z-loss:** This regularization term penalizes the magnitude of the routing logits to mitigate over-confident expert assignments and support stable gradient flow throughout training. For a batch size $B$, where each sample contains $T$ tokens, the router z-loss is formulated as:

$$\mathcal{L}_z = \lambda_z \cdot \frac{1}{B \cdot T} \sum_{b=1}^{B} \sum_{t=1}^{T} \|z_{b,t}\|_2^2,$$

where $z_{b,t} \in \mathbb{R}^N$ is the vector of raw routing logits for token $t$ in sample $b$, $\|\cdot\|_2$ denotes the $\ell_2$-norm, and $\lambda_z$ is a regularization coefficient controlling the penalty strength. This quadratic penalty, distributed over the entire batch, encourages the router to generate smoothly varying logits with lower variance, enhancing routing stability and mitigating expert collapse.

**2. Load balancing loss:** This loss promotes uniform utilization of experts across all tokens and samples, mitigating the risk of expert under-utilization or collapse. For a batch size $B$, the load balancing loss is defined as:

$$\mathcal{L}_{\text{balance}} = \lambda_b \cdot N \cdot \frac{1}{(B \cdot T)^2} \sum_{i=1}^{N} \left( \sum_{b=1}^{B} \sum_{t=1}^{T} p_{b,t,i} \right) \cdot \left( \sum_{b=1}^{B} \sum_{t=1}^{T} \mathbb{1}\left[ i \in \text{TopK}(z_{b,t}) \right] \right),$$

where $p_{b,t,i} = \frac{\exp(z_{b,t,i})}{\sum_{j=1}^{N} \exp(z_{b,t,j})}$ is the softmax probability of assigning token $t$ in sample $b$ to expert $i$. $\mathbb{1}[i \in \text{TopK}(z_{b,t})]$ is an indicator function that equals 1 if expert $i$ is among the top-$k$ selected experts for token $t$ in sample $b$, and 0 otherwise. The hyperparameter $\lambda_b$ controls the strength of this regularization term. This formulation jointly considers the *importance* of expert $i$ (measured by the sum of routing probabilities across all tokens) and the *load* (the count of tokens routed to expert $i$). The inclusion of $\mathcal{L}_{\text{balance}}$ in the final objective promotes balanced expert selection and prevents bottlenecks in expert utilization.

The total loss is defined as:
$$\mathcal{L}_{\text{total}} = \mathcal{L}_{\text{face}} + \lambda_1 \mathcal{L}_z + \lambda_2 \mathcal{L}_{\text{balance}},$$

where $\lambda_1$ and $\lambda_2$ are the weighting factor of router-z loss and load-balancing loss, respectively. This joint optimization framework allows FaceMoE to efficiently scale model capacity while dynamically specializing experts to different facial regions, thereby enhancing low-resolution face recognition performance. The FaceMoE architecture is shown in Figure 2 and the training procedure is shown in Algorithm 1.

---

**Algorithm 1** FaceMoE Training Framework

**Input:** Training samples $\{x^{(k)}, y^{(k)}\}_{k=1}^{K}$,
FaceMoE weights $\theta = \{\theta_1, \ldots, \theta_N, W_r\}$,
Experts $\{f_i(\cdot; \theta_i)\}_{i=1}^{N}$, Router Weights $W_r$
**Hyperparameters:** $\lambda, \lambda_z, \lambda_b$
**Output:** Trained FaceMoE weights $\theta$
1  **for** *each training epoch* **do**
2     **for** *each batch* $\{x_b, y_b\}_{b=1}^{B}$ **do**
3        **for** *each token* $x_{b,t}$ *in* $x_b$ **do**
4           $z_{b,t} = x_{b,t} W_r$       ▷ compute routing logits
            $(i_1, \ldots, i_k) = \text{TopK}(z_{b,t})$    ▷ select top-$k$ experts
            $w_{i_j}(x_{b,t}) = \frac{\exp(z_{b,t,i_j})}{\sum_{l=1}^{k} \exp(z_{b,t,i_l})}$    ▷ routing weights
            $p_{b,t,i} = \frac{\exp(z_{b,t,i})}{\sum_{j=1}^{N} \exp(z_{b,t,j})}$    ▷ softmax prob. for $f_i$
            $y_{b,t} = \sum_{j=1}^{k} w_{i_j}(x_{b,t}) f_{i_j}(x_{b,t})$    ▷ MoE output
5     **end**
6     $\mathcal{L}_{\text{face}} = CosFace(y_{b,t})$    ▷ face recognition loss
      $\mathcal{L}_z = \lambda_z \cdot \frac{1}{BT} \sum_{b,t} \|z_{b,t}\|_2^2$    ▷ router z-loss
      $\mathcal{L}_{\text{balance}} = \lambda_b N \frac{1}{(BT)^2} \sum_i \left( \sum_{b,t} p_{b,t,i} \right)$
      $\cdot \left( \sum_{b,t} \mathbb{1}[i \in (i_1, \ldots, i_k)] \right)$    ▷ load balancing loss
      $\mathcal{L}_{\text{total}} = \mathcal{L}_{\text{face}} + \lambda(\mathcal{L}_z + \mathcal{L}_{\text{balance}})$    ▷ total loss
      $\theta \leftarrow \texttt{Optimizer}(\theta, \nabla_\theta \mathcal{L}_{\text{total}})$    ▷ parameter update
7     **end**
8  **end**

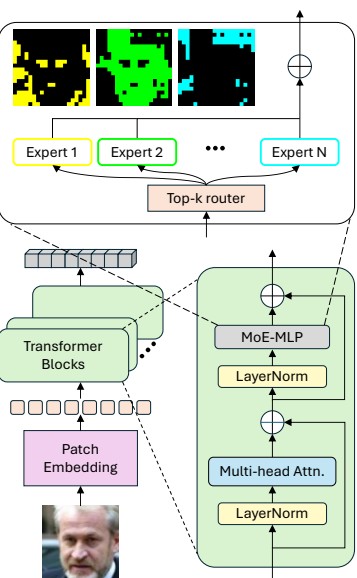

Figure 2: FaceMoE Architecture.

---

# 4 EXPERIMENTAL SETUP

**Datasets.** We use WebFace4M Zhu et al. (2021) as our pre-training dataset, which consist of approximately 4M images, with $205,990$ identities. To demonstrate the effectiveness of the proposed FaceMoE for low-resolution face recognition, we evaluate it on 3 low-resolution datasets. Further, to validate the minimal drop in pretrained performance, we also evaluate its performance on 6 high-quality datasets and 2 mixed-quality datasets. The high-quality datasets include LFW Huang et al. (2008), CFP-FP Sengupta et al. (2016), CPLFW Zheng & Deng (2018), AgeDB Moschoglou et al. (2017), CALFW Zheng et al. (2017), and CFP-FF Sengupta et al. (2016). The mixed-quality datasets are IJB-B Whitelam et al. (2017) and IJB-C Maze et al. (2018). The low-resolution datasets include TinyFace Cheng et al. (2019), IJB-S Kalka et al. (2018), and BRIAR 3.1 Cornett et al. (2023). The TinyFace Cheng et al. (2019) dataset contains $169,403$ low-resolution images spanning $5,139$ identities, with a designated training subset of $7,804$ images covering $2,570$ identities. The IJB-S Kalka et al. (2018) dataset, designed for surveillance video-based face recognition, comprises 398 videos and 202 identities. We evaluate it under *Surveillance-to-Surveillance* protocol, where "*Surveillance*" refers to footage from surveillance cameras. The BRIAR Cornett et al. (2023) training set includes $550,000$ images from 577 distinct identities. For the BRIAR evaluation, we follow Protocol 3.1 (face-included treatment), in line with prior works Jawade et al. (2024a;b). This evaluation protocol

| Method | | TAR@FAR | | |
|---|---|---|---|---|
| | | 0.01% | 0.1% | 1% |
| **Pretrained** | | | | |
| CosFace (R50) | | 22.55 | 35.43 | 52.20 |
| CosFace (ViT-B) | | 34.29 | 47.41 | 62.81 |
| CosFace (Swin-B) | | 33.77 | 45.93 | 61.17 |
| **Finetuned on BRIAR train set** | | | | |
| GAP | [ICLR 2014] | 31.70 | 40.81 | 50.76 |
| NAN | [CVPR 2017] | 34.86 | 44.96 | 54.44 |
| CosFace | [CVPR 2018] | 11.62 | 29.68 | 58.66 |
| | [BMVC 2018] | 34.84 | 45.01 | 54.25 |
| CAFace | [NeurIPS 2022] | 33.41 | 41.95 | 51.31 |
| CoNAN | [IJCB 2023] | 36.52 | 46.14 | 56.32 |
| ProxyFusion | [NeurIPS 2024] | 40.10 | 53.90 | 68.90 |
| PETAL*face* | [WaCV 2025] | 35.12 | 55.35 | 75.43 |
| **FaceMoE** | | **42.36** | **61.47** | **81.27** |

Table 1: Results on BRIAR Protocol 3.1.

| Method | | TPIR@FPIR | Rank Retrieval | |
|---|---|---|---|---|
| | | 1% | Rank-1 | Rank-5 |
| **Pretrained** | | | | |
| CosFace (R50) | | 3.67 | 33.62 | 49.40 |
| CosFace (ViT-B) | | 2.58 | 25.76 | 40.69 |
| CosFace (Swin-B) | | 2.11 | 22.52 | 37.97 |
| **Finetuned on BRIAR train set** | | | | |
| CosFace | [CVPR 2018] | 1.72 | 16.44 | 31.58 |
| PFE | [CVPR 2019] | 0.84 | 9.20 | 20.82 |
| RSA | [ICCV 2019] | 0.75 | 16.82 | 31.80 |
| MARN | [ICCVW 2019] | 0.19 | 22.25 | 34.16 |
| ArcFace | [CVPR 2019] | 5.32 | 32.13 | 46.67 |
| CFAN | [IJCB 2019] | 5.79 | 31.66 | 45.59 |
| CurricularFace | [CVPR 2020] | 2.53 | 19.54 | 32.80 |
| AdaFace | [CVPR 2022] | 4.96 | 35.05 | 48.22 |
| CAFace | [NeurIPS 2022] | 8.78 | 36.51 | 49.59 |
| PETAL*face* | [WaCV 2025] | 12.25 | 38.32 | 51.50 |
| **FaceMoE** | | **14.85** | **44.81** | **56.12** |

Table 2: Results on IJB-S (Surv. to Surv.).

features a gallery of $86,958$ controlled images representing $615$ identities and a probe set comprising $5,435$ clips from $260$ identities.

**Evaluation Setup and Metrics.** We organize our experiments into two protocols to comprehensively evaluate FaceMoE across a variety of scenarios. In **Protocol-1**, we pre-train FaceMoE on the Web-Face4M Zhu et al. (2021), finetune it on the challenging low-resolution BRIAR Cornett et al. (2023) dataset, and evaluate its performance using BRIAR Protocol 3.1, demonstrating the effectiveness of FaceMoE for low-resolution face recognition. We also test the model on IJB-S Kalka et al. (2018) which is another challenging video-surveillance dataset to show its out-of-distribution performance. In **Protocol-2**, we finetune our model on TinyFace Cheng et al. (2019) and evaluate it on its test set. With this protocol, we aim to highlight the capability of FaceMoE to adapt to low-resolution datasets while maintaining performance on high-resolution and mixed-quality datasets. We evaluate the models on high-resolution and mixed-quality datasets using 1:1 verification accuracy and TAR@FAR across various thresholds. For TinyFace, we apply rank retrieval metrics at Rank-1, Rank-5, and Rank-10. On the BRIAR dataset, we report both TAR@FAR at different thresholds and closed-set rank retrieval at Rank-1, Rank-5, and Rank-20. For IJB-S, we evaluate open-set performance using TPIR@FPIR = 1% and 10%, along with closed-set rank retrieval at Rank-1, Rank-5, and Rank-10.

**Implementation Details.** We train FaceMoE on WebFace4M with a batch size of 128 per GPU for 26 epochs, using AdamW (weight decay $5 \times 10^{-2}$) and a Polynomial LR scheduler with 1 warmup epoch and initial LR $10^{-3}$. Fine-tuning is done on TinyFace and BRIAR in two stages: linear probing and full fine-tuning. For TinyFace, linear probing runs 10 epochs (2 warmup) at LR $10^{-3}$, batch size 16; full fine-tuning runs 40 epochs (4 warmup) at LR $10^{-4}$, batch size 8. For BRIAR, both stages run 20 epochs (2 warmup), with LRs $10^{-3}$ and $5 \times 10^{-6}$, batch sizes 64 and 8. Training uses face recognition loss, router z-loss, and load balancing loss with $\lambda_1 = 10$, $\lambda_2 = 10$, $\lambda_z = 1$, $\lambda_b = 1$. The $\lambda_1$ and $\lambda_2$ values scale the auxiliary loss terms so that their magnitudes are comparable to the main face recognition loss, ensuring stable optimization without either term dominating training. The best results are obtained with 3 experts ($N = 3$) and 2 active experts per token ($k = 2$). All experiments use PyTorch on eight NVIDIA A6000 GPUs (48GB). Additional details are provided in the appendix.

## 5 RESULTS AND ANALYSIS

**We encourage the reader to have a look at the additional results, analysis and ablation studies discussed in Section B**

**Results on Protocol 1:** The results for Protocol 1 are summarized in Table 1 and Table 2. The pretrained transformer backbones ViT-B and Swin-B show superior performance than ResNet-50, however these models are not finetuned on low-resolution datasets and perform poorly compared to finetuned methods. Traditional feature aggregation methods such as GAP Lin et al. (2013), NAN Yang et al. (2017), MCN Xie & Zisserman (2018), CAFace Kim et al. (2022b), and CoNAN Jawade et al. (2024a) yield incremental improvements, but remain limited in their ability to extract discriminative identity features from degraded probe images, as they use a feature encoder with single FFN and focus on selecting relevant frames with sufficient identity information. However, our method aims

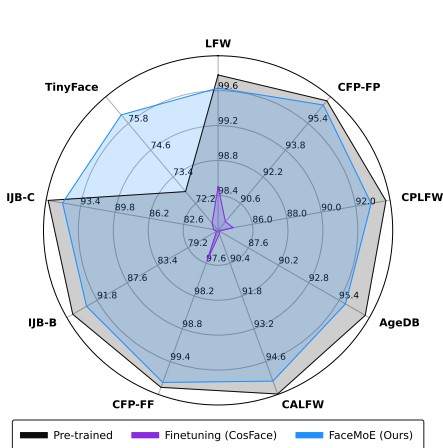

Figure 3: FaceMoE incurs minimal performance drop on HR and mixed-quality datasets, effectively extracting features from HR gallery and LR probe.

| Method | Arch. | Data | Rank-1 | Rank-5 | Rank-10 |
|---|---|---|---|---|---|
| **Pretrained** | | | | | |
| URL | R-100 | MS1MV2 | 63.89 | 68.67 | – |
| CurricularFace | R-100 | MS1MV2 | 63.68 | 67.65 | – |
| CosFace | R-50 | WF4M | 72.71 | 76.36 | 78.99 |
| ArcFace | R-50 | WF4M | 73.04 | 76.85 | 79.45 |
| AdaFace | R-50 | WF4M | 73.49 | 76.60 | 79.07 |
| CosFace | ViT-B | WF4M | 73.57 | 76.95 | 78.94 |
| ArcFace | ViT-B | WF4M | 72.74 | 76.28 | 78.13 |
| AdaFace | ViT-B | WF4M | 74.03 | 77.22 | 79.37 |
| CosFace | Swin-B | WF4M | 72.74 | 76.79 | 79.18 |
| ArcFace | Swin-B | WF4M | 73.31 | 76.68 | 79.23 |
| AdaFace | Swin-B | WF4M | 74.40 | 77.62 | 79.51 |
| KP-RPE | ViT-B | WF4M | 75.80 | 78.49 | – |
| **Finetuned on TinyFace** | | | | | |
| CosFace | Swin-B | WF4M | 71.32 | 76.42 | 79.45 |
| ArcFace | Swin-B | WF4M | 71.11 | 76.63 | 79.96 |
| PETAL*face* | Swin-B | WF4M | 75.45 | 79.05 | 81.19 |
| **FaceMoE (Ours)** | Swin-B | WF4M | **76.18** | **79.69** | **81.75** |

Figure 4: Results on TinyFace. Pre-trained models when finetuned on TinyFace dataset results in performance drop. FaceMoE achieves SOTA performance and is capable of adapting to low-resolution dataset with minimal performance drop in HQ and mixed-quality dataset.

to improve the identity extraction of all the frames by improving the feature extractor itself. Recent methods, ProxyFusion and PETAL*face*, achieve a TAR@FAR of 40.10, 53.90, 68.90 & 35.12, 55.35, 75.43 at thresholds 0.01%, 0.1% & 1%, resp.

Our proposed FaceMoE achieves the highest performance across all thresholds with 42.36%, 61.47%, and 81.27% TAR at 0.01%, 0.1%, and 1% FAR, respectively. The superior performance of FaceMoE can be attributed to its *resolution-aware feature extraction* enabled by specialized experts. Each expert is implicitly trained to focus on distinct semantic regions of the face, such as edges, contours, or landmark regions, enabling dynamic adaptation to severely degraded probe images. This capability is especially valuable in low-resolution scenarios, where identity information is limited and often confined to localized regions. In such cases, key identity discriminative features, such as the eyes, nose, or mouth may be occluded, blurred, or affected by extreme lighting conditions. FaceMoE addresses this by allotting specialized semantic experts to other informative regions, enabling a more robust and comprehensive identity representation. This enhanced feature extraction from low-resolution probes directly contributes to superior feature aggregation, resulting in state-of-the-art performance for low-resolution face recognition on the BRIAR dataset. Table 2 reports the generalization performance on the IJB-S dataset under *Surveillance-to-Surveillance* protocol. We observe similar trends, with FaceMoE outperforming all prior methods by a significant margin. FaceMoE achieves 14.85% TPIR at 1% FPIR, along with 44.81% and 56.12% Rank-1 and Rank-5 retrieval accuracies, respectively. The *resolution-aware feature extraction* and expert specialization effectively handle the extreme variability and degradation inherent in surveillance footage, extracting identity features from limited and inconsistent information across frames. This enhanced feature extraction leads to robust identity recognition under the most challenging low-resolution conditions.

**Results on Protocol 2:** The results for Protocol 2 are shown in Figure 4 and 3. Finetuning pretrained models such as CosFace Wang et al. (2018) and ArcFace Deng et al. (2019) on TinyFace leads to a drop in performance not only on the LR dataset but also on the mixed-quality and HR datasets. This degradation is primarily due to catastrophic forgetting, as these models lack mechanisms to effectively adapt to low-resolution data while retaining the discriminative features learned during pretraining. This effect can also be observed in Table 1 and Table 2, where finetuned CosFace shows a significant performance drop on BRIAR Protocol 3.1 and IJB-S compared to pretrained CosFace. In contrast, FaceMoE establishes a new state-of-the-art on TinyFace with 76.18%, 79.69%, and 81.75% Rank-1, Rank-5, and Rank-10 retrieval accuracy, respectively, with a minimal drop in performance on the HR and mixed quality datasets as illustrated in Figure 3.

The superior performance of FaceMoE can be attributed to its unique architectural design, which leverages multiple sparse FFN experts to facilitate effective adaptation to low-resolution datasets, while incurring minimal performance drop on high-resolution and mixed-quality datasets. The

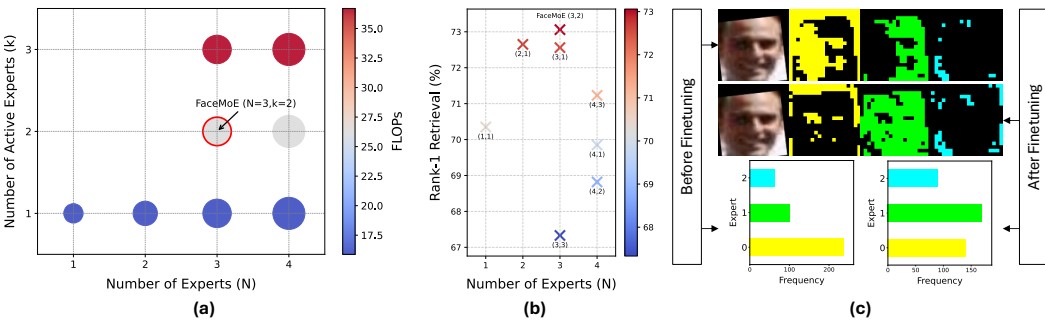

Figure 5: (a) Computational trade-off analysis across different MoE configurations (Bubble size $\propto$ #Parameters). (b) Impact of $N$ and $k$ on performance, evaluated on the BRIAR dataset. (c) FaceMoE expert activation maps and token assignment histograms before and after fine-tuning on a low-resolution dataset. The updated token assignments indicate *resolution-aware feature extraction*, while the semantically coherent expert activation maps demonstrate stable convergence.

top-$k$ router renders the network modular and sparsely activated, restricting weight updates during finetuning to only a subset of experts. As a result, the model avoids *catastrophic forgetting* as observed in traditional models. During finetuning of FaceMoE, the model exhibits a phenomenon known as selective drift Rypeść et al. (2024), where certain experts adapt specifically to the low-resolution dataset, while others retain the pretrained knowledge. As shown in Figure 5(c), expert 2's focus remains largely consistent before and after finetuning, focusing on broader facial shapes, indicating the preservation of pretrained semantic knowledge. However, token assignment changes significantly during finetuning: before finetuning, expert 0 was predominantly utilized, whereas after finetuning, expert 1 becomes more active. This shift highlights FaceMoE's resolution-aware capability and its dynamic utilization of experts based on input resolution. The expert activation maps after finetuning display more semantically coherent and well-defined regions, showcasing the efficacy of employing multiple FFN experts in conjunction with a top-$k$ router for stable adaptation to low-resolution data. FaceMoE's ability to adapt to low-resolution data while preserving pretrained knowledge enables effective feature extraction across high-resolution gallery and low-resolution probe domains.

**Impact of $N$ and $k$ on Performance:** We perform an ablation study to investigate the effect of the number of experts ($N$) and the number of active experts per token ($k$) on model performance. Figure 5(b) shows the Rank-1 retrieval accuracy on the BRIAR dataset for different $(N, k)$ configurations. We observe that both under-parameterization and over-parameterization can adversely impact performance. A low number of experts ($N = 1$) limits the model's capacity to specialize across facial regions, resulting in sub-optimal performance (70.2%). On the other hand, increasing the number of experts excessively ($N = 4$) introduces routing instability and model fragmentation, leading to degraded performance across multiple $k$ settings. Our best performance is achieved with $N = 3$ experts and $k = 2$ active experts per token, corresponding to the FaceMoE configuration, which achieves 73.1% Rank-1 retrieval. This setting strikes an effective balance between model capacity and routing stability, providing sufficient expert diversity to allow specialization across semantic regions (e.g., hair, landmarks, textures), while avoiding excessive fragmentation of the feature space.

**Computational Analysis:** We study the computational cost of different $(N, k)$ configurations. Figure 5(a) shows the FLOPs for various combinations of number of experts $N$ and active experts per token $k$. As expected, computational cost scales with $k$, since more experts are evaluated per token. Importantly, for fixed $k$, the parameter count remains constant regardless of $N$, as only $k$ experts contribute to the forward pass. For example, with $k = 2$, both $(N = 3, k = 2)$ and $(N = 4, k = 2)$ have the same number of active parameters with 26.29 GFLOPs, despite differing in total experts. The optimal configuration for FaceMoE is $(N = 3, k = 2)$, achieving a favorable trade-off between model capacity and computational cost. This results in a moderate 26.29 GFLOPs, offering a 2.17× increase in capacity over the standard Swin-B backbone (15.88 GFLOPs) with only a 1.66× increase in FLOPs. This validates the efficiency of sparsely activated experts, enabling the model to significantly boost its representation power while maintaining practical inference cost.

## 6 CONCLUSION

In this work, we present FaceMoE, a novel transformer-based architecture enhanced with a Mixture of Experts mechanism to address persistent challenges in low-resolution face recognition. We

incorporate multiple FFN experts and a top-$k$ router, enabling the experts to specialize in different semantic regions of the face. The proposed framework enhances the discriminative power of feature extraction under severe image degradations, and the presence of multiple FFN experts ensures stable finetuning with minimal performance loss on high-resolution and mixed-quality datasets. Extensive evaluations across eleven diverse benchmarks, including challenging low-resolution datasets such as TinyFace, IJB-S, and BRIAR, demonstrate that FaceMoE consistently outperforms existing methods, establishing new SOTA performance in low-resolution face recognition.

## ETHICS STATEMENT

In this research, we have carefully addressed the ethical implications surrounding face recognition technology, particularly focusing on issues of privacy, surveillance, and potential biases. Our model was trained on publicly available datasets: WebFace4M and WebFace12M Zhu et al. (2021), acquired through signing the official license agreement. For benchmarking, we utilized IJB-B Whitelam et al. (2017), IJB-C Maze et al. (2018), IJB-S Kalka et al. (2018), BRIAR Cornett et al. (2023), and TinyFace Cheng et al. (2019), which contain diverse, mixed-quality, and low-resolution images from real-world settings. These datasets were obtained through official repositories and websites, ensuring adherence to ethical standards. Informed consent for publication was acquired for all subjects depicted in the paper, supporting ethical data use.

This research offers significant benefits within authorized security contexts, where accurate low-resolution face recognition enhances identification capabilities in challenging environments. When applied responsibly, these advancements contribute to security and enable legitimate monitoring efforts. Importantly, the model's design and training process adhere to standards that do not introduce risks beyond those inherent in traditional face recognition systems. However, we acknowledge the potential for misuse in unauthorized surveillance, profiling, or privacy infringements if deployed outside controlled, ethical frameworks. Our work aims to support face recognition for responsible use within authorized security settings, while recognizing that unintended applications or misinterpretations could lead to societal issues, such as privacy erosion or biased treatment of certain groups. By proactively addressing these considerations, we seek to mitigate risks associated with the model's deployment and advocate for ethical oversight to prevent misuse.

Ethical considerations for human subjects and data usage were fully respected. This research relies solely on existing datasets and no new consent was required. These datasets are approved for research use, ensuring adherence to ethical data standards. No individuals were recruited which eliminates the need for compensation. The datasets do not predominantly include vulnerable populations, such as minors, elderly individuals, or other at-risk groups, instead representing a standard demographic spectrum. Given our commitment to ethical standards, this research presents minimal risk to individuals while advancing low-resolution face recognition technology.

## REPRODUCIBILITY STATEMENT

We ensure the reproducibility of our work by providing full implementation details, including the FaceMoE architecture, training framework, and evaluation protocols. Our method is implemented in PyTorch with widely available libraries, and we will release code, pretrained models, and fine-tuning scripts upon acceptance. All datasets used (WebFace4M, TinyFace, IJB-B, IJB-C, IJB-S, and BRIAR) are publicly available through official repositories or license agreements, and we strictly follow their established evaluation protocols (e.g., BRIAR Protocol 3.1, IJB-S Surveillance-to-Surveillance). Hyperparameters such as learning rates, batch sizes, epochs, and warm-up schedules, as well as optimizer details (AdamW, weight decay, polynomial LR scheduler), are fully described in Section 4, with pseudocode provided in Algorithm 1.

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

## APPENDIX

As part of the appendix, we present the following as an extension to the ones shown in the paper:

## A    RELATED WORK: MOE IN FACE ANALYSIS TASKS

AMEL Chen et al. (2025) combines a shared expert with low-rank adapted attack-specific experts and dynamically aggregates them to improve robustness to post-processing distortions in audio spoofing detection. MoE-FFD Kong et al. (2025) fuses transformer's global features with CNN-style local priors and uses a gating scheme to dynamically select the most relevant forgery expert, boosting generalization across face forgery types. However, the proposed FaceMoE differs from AMEL Chen et al. (2025) and MoE-FFD Kong et al. (2025) in several aspects, along with the fact that they target different tasks, such as:

- **Expert Integration:** *FaceMoE* incorporates multiple FFN-based experts within the MLP layers of a transformer encoder, enabling semantic specialization for different facial regions. *AMEL* introduces Domain-Specific Experts (DSEs), which are lightweight residual blocks appended to a shared CNN backbone, each modeling features from a specific source domain. *MoE-FFD* adopts parameter-efficient experts using LoRA and Adapter modules, allowing expert injection without modifying the backbone, and is optimized for face forgery detection.

- **Routing Strategy:** *FaceMoE* employs a learned top-$k$ router at the token level, directing face patches to a subset of specialized experts based on resolution and semantic cues. *AMEL* uses Dynamic Expert Aggregation (DEA) at the sample level, computing soft aggregation weights across domain experts based on domain similarity. *MoE-FFD* utilizes a top-1 gating mechanism to assign each input to the most relevant forgery expert, with routing performed at the sample level for efficiency.

- **Training Paradigm:** *FaceMoE* enables full fine-tuning of the transformer model, complemented by auxiliary losses (router z-loss and load-balancing) to promote expert specialization and training stability. *AMEL* is trained using a meta-learning strategy that simulates domain shifts across source domains; it combines standard classification and depth supervision with a feature consistency loss. *MoE-FFD* adopts a parameter-efficient fine-tuning (PEFT) strategy, keeping the backbone frozen while training only the inserted expert modules, thereby reducing training overhead.

# B ADDITIONAL RESULTS, ANALYSIS AND ABLATION STUDIES

## B.1 EXPERT SPECIALIZATION

| Facial Region | Expert 0 | Expert 1 | Expert 2 | Dominant Expert |
|---|---|---|---|---|
| Eyes | **76.2** | 12.4 | 11.4 | Expert 0 |
| Nose | **68.5** | 15.7 | 15.8 | Expert 0 |
| Mouth | **61.0** | 22.1 | 16.9 | Expert 0 |
| Forehead | 13.2 | **71.3** | 15.5 | Expert 1 |
| Cheeks | 12.5 | **69.8** | 17.7 | Expert 1 |
| Hair / Background | 20.3 | 14.5 | **65.2** | Expert 2 |
| Chin / Jawline | 25.6 | 18.4 | **56.0** | Expert 2 |

Table B.1: Routing frequency (%) of each expert across different semantic facial regions. Results averaged over 10,000 samples.

| Facial Region | Expert 0 | Expert 1 | Expert 2 | Dominant Expert |
|---|---|---|---|---|
| Eyes | **81.7** | 10.6 | 7.7 | Expert 0 |
| Nose | **78.1** | 12.5 | 9.4 | Expert 0 |
| Mouth | **71.6** | 16.0 | 12.4 | Expert 0 |
| Forehead | 49.3 | **31.1** | 19.6 | Expert 1 |
| Cheeks | 52.5 | **27.8** | 19.7 | Expert 1 |
| Hair / Background | 40.2 | 17.3 | **42.5** | Expert 2 |
| Chin / Jawline | 38.9 | 18.7 | **42.4** | Expert 2 |

Table B.2: Pre-finetune Routing frequency (%) of each expert across different semantic facial regions. Results averaged over 10,000 samples.

To strengthen our claim regarding expert specialization, we conducted an additional analysis quantifying the consistency and evolution of expert assignments across facial regions. We computed the routing frequency of each expert before and after finetuning on a sample of 10,000 face images. Table B.2 reports the routing behavior of the pre-finetuned model, while Table B.1 shows the corresponding results after finetuning. The comparison reveals two important trends:

**Finetuning strengthens and sharpens spatial specialization.** Before finetuning, the router exhibits only weak spatial bias: Expert 0 is mildly preferred for central regions (eyes, nose, mouth), Expert 1 receives a moderate share of tokens from the forehead and cheeks, and Expert 2 shows a slight preference for peripheral or high-frequency regions such as hair and the jawline. However, these tendencies remain relatively diffuse, as reflected by the more evenly distributed routing frequencies across experts.

After finetuning, these patterns become markedly more pronounced. Expert 0 becomes the dominant processor for identity-rich central regions (eyes, nose, mouth), consistently exceeding 60-76% routing frequency. Expert 1 specializes in smoother, low-frequency regions such as the forehead and cheeks, each receiving over 69-71% assignments. Expert 2 emerges as the primary expert for high-frequency or peripheral structures, including hair, background, and the jawline, with routing frequencies between 56-65%. These results indicate that finetuning drives the router toward strong and interpretable spatial specialization.

**Experts become more complementary and disentangled.** A direct comparison of the pre- and post-finetune distributions shows that finetuning reduces expert overlap and increases region-specific dominance. Whereas the pre-finetuned model routes many regions across experts in a relatively mixed manner (e.g., cheeks: 52.5/27.8/19.7), the finetuned model exhibits clear expert preferences (e.g., cheeks: 12.5/69.8/17.7). This shift demonstrates that the router learns to assign experts based on semantic and frequency characteristics, yielding complementary specialization across landmark, low-frequency, and high-frequency regions.

## B.2 EXPERT SPECIALIZATION MECHANISM IN FACEMOE

In this section, we provide deeper insight into the mechanism through which the experts in FaceMoE specialize in distinct facial regions. While the main paper introduces the motivation behind incorporating sparse FFN experts, here we examine how this specialization implicitly emerges during training, how it is reinforced by the top-$k$ router, and how it manifests both quantitatively and qualitatively.

FaceMoE integrates a sparse mixture-of-experts (MoE) layer within each MLP block of the transformer. Unlike a dense FFN, which applies the same transformation to all tokens, the MoE layer enables *conditional computation*, allowing each token to be processed by only a subset of experts. This design naturally promotes expert specialization. Facial tokens exhibit diverse semantic and frequency characteristics, such as high-frequency regions (edges, contours, hair boundaries), smooth low-frequency regions (cheeks, forehead), and structured landmark regions (eyes, nose, mouth). Since the router computes routing logits through a linear projection of token embeddings, tokens from these different regions generate *distinct* routing patterns even early in training. This asymmetry initiates the specialization process.

A positive feedback loop then emerges: tokens from a specific region (e.g., the eyes) initially receive slightly higher routing logits for a particular expert, leading them to be repeatedly routed to that expert. As a result, the expert's parameters gradually specialize to model the statistical patterns characteristic of those tokens. In parallel, the router learns to reinforce these region–expert correspondences. This dynamic ultimately produces experts that specialize in semantically coherent facial subsets.

**Conditional Routing Behavior** For the default configuration with 3 experts and top-2 routing, we observe that routing probabilities converge to region-consistent patterns. Empirically:

$$P(E_i \mid R_t = r) \gg P(E_i \mid R_t \neq r), \qquad r \in \{\text{high-frequency}, \text{low-frequency}, \text{landmarks}\}.$$

indicating that each expert becomes the preferred destination for a specific category of tokens.

**Quantitative Evidence of Specialization** Appendix B.1 includes statistics of token-assignment distribution showing that:

- routing variance decreases over training,
- each expert receives consistent token subsets,
- spatial patterns on the face correspond to stable expert clusters.

**Qualitative Evidence via Activation Maps** Activation maps provided in Appendix D demonstrate:

- Spatial consistency: each expert highlights distinct facial zones;
- Semantic coherence: landmark-oriented experts focus on eyes/nose/mouth;
- Resolution-aware specialization: LR images trigger higher reliance on experts specializing in coarse structural cues.

## B.3 RESOLUTION ABLATION

We conduct a resolution ablation study by varying the resolution of the test images and observe only a minimal drop in performance across resolutions as shown in Table B.3. This result reinforces FaceMoE's capability to effectively handle inputs of varying resolutions.

## B.4 BIAS ANALYSIS

We quantify the bias implications of our mixture-of-experts based FaceMoE architecture compared to the baseline to showcase that FaceMoE exhibits minimal bias. We conducted further experiments on LFW, CFP-FF, and AgeDB datasets. We used FaceXFormer Narayan et al. (2024) to infer age (0–19, 20–39, 40–59, 60+), gender (male, female), and race (Black, Latino/Hispanic, Middle Eastern, Asian, White) labels. To evaluate fairness, we adopted the Selective Ratio (SeR) and Degree of Bias (DoB) as our metrics.

| Dataset | 8x8 | 10x10 | 12x12 | 16x16 | 32x32 | 48x48 | 64x64 | 96x96 |
|---------|------|-------|-------|-------|-------|-------|-------|-------|
| LFW | 80.75 | 86.98 | 91.71 | 96.06 | 99.61 | 99.68 | 99.68 | 99.73 |
| CFP-FP | 62.38 | 68.45 | 72.94 | 80.87 | 94.75 | 96.22 | 96.57 | 96.72 |
| CPLFW | 65.20 | 71.33 | 77.18 | 82.26 | 92.35 | 93.01 | 93.23 | 93.28 |
| AgeDB-30 | 56.23 | 59.13 | 63.48 | 70.51 | 92.53 | 95.48 | 96.06 | 96.25 |
| CALFW | 63.18 | 68.58 | 74.66 | 80.31 | 93.75 | 94.76 | 95.10 | 95.43 |
| CFP-FF | 73.24 | 78.71 | 83.44 | 90.32 | 99.02 | 99.68 | 99.67 | 99.65 |

Table B.3: Accuracy (%) across different image resolutions on various datasets.

The results, summarized in Table B.4, demonstrate that FaceMoE not only achieves superior performance but also results in a fairer model with reduced bias across age, gender, and racial attributes compared to the baseline.

| Dataset | Model | Age | | Gender | | Race | |
|---------|-------|-----|-----|--------|-----|------|-----|
| | | SeR | DoB | SeR | DoB | SeR | DoB |
| LFW | Swin-B | 0.95 | 2.16 | 0.99 | 0.25 | 0.80 | 8.07 |
| | FaceMoE | 0.95 | 2.20 | 0.99 | 0.07 | 0.84 | 6.30 |
| CFP FF | Swin-B | 0.93 | 3.29 | 0.99 | 0.27 | 0.86 | 5.52 |
| | FaceMoE | 0.93 | 3.28 | 0.99 | 0.25 | 0.86 | 5.54 |
| AgeDB | Swin-B | 0.99 | 0.34 | 0.99 | 0.15 | 0.77 | 9.26 |
| | FaceMoE | 0.99 | 0.28 | 0.99 | 0.12 | 0.77 | 9.87 |

Table B.4: Performance comparison of Swin-B and FaceMoE across different datasets for Age, Gender, and Race attributes.

The intrinsic reason behind FaceMoE's improved fairness lies in its mixture-of-experts design, which encourages different experts to specialize in complementary facial regions and frequency patterns. This specialization allows the router to dynamically select the most informative experts for each input, particularly beneficial when demographic groups differ in blur level, pose variation, skin texture, or age-related changes. Consequently, the model avoids over-reliance on any single facial attribute that may be demographically sensitive, leading to more stable SeR/DoB scores across groups. In practice, FaceMoE appears fairer because (1) sparse expert activation mitigates biased drift during fine-tuning, and (2) expert diversity distributes representational responsibility across multiple specialized pathways rather than amplifying group-specific biases.

## B.5 COMPARISON WITH OTHER MOE VARIANTS

We conducted ablations on multiple MoE configurations to evaluate their effectiveness for low-resolution face recognition as shown in Table B.5:

- **Shared MoE:** A single shared MLP expert activated for all tokens. This limits specialization and makes the model overly rigid when adapting to low-resolution data, leading to degraded performance.

- **LoRA FFN:** Uses LoRA experts instead of dense FFNs. While lightweight, it lacks sufficient representational capacity and is prone to catastrophic forgetting.

- **LoRA FFN + Attn:** Extends LoRA to Q, K, V projections. Although slightly better, it still lacks expressiveness for resolution-aware specialization.

- **FaceMoE (Ours):** Incorporates multiple full-capacity FFN experts in transformer MLP layers, combined with a token-wise top-$k$ router. This enables spatially-aware, resolution-sensitive expert activation, leading to significantly better performance.

| Method | Rank-1 | Rank-5 | Rank-10 |
|---|---|---|---|
| Shared MoE | 62.71 | 69.39 | 73.92 |
| LoRA FFN | 43.61 | 52.62 | 59.81 |
| LoRA FFN + Attn | 44.98 | 53.37 | 60.48 |
| **FaceMoE (Ours)** | **76.18** | **79.69** | **81.75** |

Table B.5: Performance comparison of different MoE variants on TinyFace.

### B.6 IMPACT OF EACH COMPONENT

We provide an ablation study in Table B.6 to assess the impact of each component in FaceMoE. We see a drop in Rank-1 accuracy from 76.18% to 75.40%, if we remove the top-k router, highlighting the importance of dynamic token routing. We see a further reduction in performance to (75.10%), if we exclude the MoE module, confirming the benefit of expert specialization. Finally, omitting the auxiliary losses results in the lowest accuracy of (74.94%), underscoring their role in stabilizing routing and balancing expert utilization. These results demonstrate that all components contribute meaningfully to the overall performance. Please note that we cannot report a configuration with MoE, Aux Loss but without a top-$k$ router, as the auxiliary loss depends on the logits produced by the top-$k$ router and cannot function without them.

| MoE | Top-$k$ Router | Aux Loss | Rank-1 | Rank-5 | Rank-10 |
|---|---|---|---|---|---|
| ✗ | ✗ | ✗ | 75.10 | 78.16 | 80.20 |
| ✓ | ✗ | ✗ | 75.40 | 78.46 | 80.63 |
| ✓ | ✓ | ✗ | 74.94 | 77.92 | 79.90 |
| ✓ | ✓ | ✓ | 76.18 | 79.69 | 81.75 |

Table B.6: Ablation study showing the impact of top-k router, MoE, and auxiliary loss on FaceMoE performance. Results are shown on TinyFace dataset

### B.7 LARGE $N$ AND RANDOM EXPERT ASSIGNMENT

To evaluate whether the observed performance gains stem from meaningful expert specialization or simply increased model capacity, we conducted controlled experiments with (a) random expert assignment and (b) increased number of experts (N = 8). The results are shown in Table B.7.

As evident, using random expert assignment (i.e., bypassing learned routing) results in a performance drop across all metrics (e.g., Rank-1: 75.40 vs. 76.18), suggesting that the learned routing mechanism does contribute meaningfully to the model's discriminative ability. More notably, increasing the number of experts to N = 8 leads to training collapse (Rank-1: 2.31), highlighting that larger expert sets can destabilize training without proper balancing, as discussed in Figure 5(a)(b)). This instability stems from expert under-utilization and routing noise.

These results collectively support our claim that expert specialization, when properly routed and regularized, is fundamental to the model's performance, not merely a byproduct of added parameters.

| | Rank-1 | Rank-5 | Rank-10 |
|---|---|---|---|
| FaceMoE | 76.18 | 79.69 | 81.75 |
| Random Expert Assignment | 75.40 | 78.46 | 80.63 |
| Large N (N=8) | 2.31 | 3.82 | 6.04 |

Table B.7: Performance comparison of Random Experts and Large N on TinyFace dataset.

### B.8 BACKBONE ABLATION

To evaluate the backbone-agnostic nature of FaceMoE, we conduct experiments using both the standard Vision Transformer (ViT-B) and the hierarchical Swin Transformer (Swin-B). Table B.8 presents performance results across four challenging benchmarks: IJB-B and IJB-C (TAR at FAR = $10^{-4}$),

TinyFace (Rank-1), and BRIAR Protocol 3.1 (Rank-1/5/20). The results lead to four key observations. First, FaceMoE integrates seamlessly with both ViT-B and Swin-B architectures without requiring any architecture-specific modifications, highlighting its generality. Second, FaceMoE-equipped models retain performance on IJB-B and IJB-C that is comparable to the ViT-B baseline, demonstrating that the Mixture-of-Experts routing mechanism preserves the generalizable features learned during pretraining. Third, FaceMoE consistently improves performance on difficult benchmarks, including an approximately 2.3% absolute increase in Rank-1 accuracy on TinyFace and a notable 15.8% gain on BRIAR Protocol 3.1 (Rank-1). Finally, combining FaceMoE with the hierarchical Swin-B backbone yields further performance improvements, particularly under stringent evaluation settings, such as a 1.72% increase in Rank-1 accuracy on BRIAR. These findings collectively confirm that FaceMoE is inherently backbone-agnostic, maintains pretrained discriminative capacity, and significantly enhances robustness in low-FAR and low-resolution face recognition scenarios.

| Backbone | IJBB | IJBC | TinyFace | BRIAR Protocol 3.1 | | |
|---|---|---|---|---|---|---|
| | e-4 | e-4 | Rank-1 | Rank-1 | Rank-5 | Rank-20 |
| ViT-B | 95.18 | 96.87 | 73.57 | 55.59 | 63.44 | 72.76 |
| ViT-B (FaceMoE) | 89.75 | 92.08 | 75.85 | 71.34 | 80.24 | 89.20 |
| Swin-B (FaceMoE) | 93.27 | 95.28 | 76.18 | 73.06 | 82.18 | 89.03 |

Table B.8: Results of FaceMoE with ViT-B backbone on IJBB, IJBC, TinyFace, and BRIAR Protocol 3.1. FaceMoE works for all kind of transformer backbones.

## B.9 PERFORMANCE WITH DATA SCALING

| Pretraining Dataset | IJBB | IJBC | TinyFace | BRIAR Protocol 3.1 | | |
|---|---|---|---|---|---|---|
| | e-4 | e-4 | Rank-1 | Rank-1 | Rank-5 | Rank-20 |
| WebFace4M | 93.27 | 95.28 | 76.18 | 73.06 | 82.18 | 89.03 |
| **WebFace12M** | **93.77** | **95.66** | **76.42** | **74.77** | **83.36** | **90.56** |

Table B.9: Performance of FaceMoE improves with increase in pre-training dataset size.

When we increase the size of the pre-training dataset from WebFace4M to WebFace12M, FaceMoE's performance consistently improves across a spectrum of face recognition benchmarks. On the IJBB protocol at a FAR of $1e^{-4}$ (after fine-tuning on TinyFace), we observe a gain from 93.27% to 93.77%. A similar trend holds on IJBC (also after TinyFace fine-tuning), where accuracy at the same operating point increases by 0.38, from 95.28% to 95.66%. Even on the challenging TinyFace dataset, where both pre-trained models are further fine-tuned on TinyFace, the Rank-1 accuracy climbs from 76.18% to 76.42%, demonstrating that additional data yields measurable benefits under difficult, low-resolution conditions. The gains are most pronounced on the BRIAR Protocol 3.1 benchmarks (after BRIAR fine-tuning), with Rank-1 accuracy improving by 1.71 (from 73.06% to 74.77%), Rank-5 by 1.18 (from 82.18% to 83.36%), and Rank-20 by 1.53 (from 89.03% to 90.56%). These results not only confirm that FaceMoE continues to harness extra data to push its recognition capabilities forward, but also illustrate strong preservation of pre-trained knowledge through successive fine-tuning stages.

All data scaling results are shown in Table B.9, where IJBB and IJBC results are reported after fine-tuning on TinyFace; the TinyFace results likewise follow TinyFace fine-tuning; and the BRIAR Protocol 3.1 results are after BRIAR fine-tuning. When the pre-training dataset is increased from WebFace4M to WebFace12M, FaceMoE's performance improves uniformly across all benchmarks. On IJBB at a FAR of $1 \times 10^{-4}$, the TAR rises from 93.27% to 93.77% (+0.50). Similarly, on IJBC under the same operating point, TAR increases by 0.38, from 95.28% to 95.66%. On TinyFace, Rank-1 accuracy climbs from 76.18% to 76.42% (+0.24), demonstrating benefits even under low-resolution conditions. The most substantial gains appear on BRIAR Protocol 3.1: Rank-1 improves by 1.71 (from 73.06% to 74.77%), Rank-5 by 1.18 (from 82.18% to 83.36%), and Rank-20 by 1.53 (from 89.03% to 90.56%). These results confirm that scaling the pre-training data both enhances FaceMoE's

recognition accuracy and preserves its learned representations after fine-tuning on low-resolution face recognition dataset.

Several architectural and training factors contribute to the successful scaling of data. First, the mixture-of-experts design enables conditional computation. Although the overall model capacity increases with the addition of more experts, each input activates only a small subset of them. This means that tripling the dataset size does not significantly increase the computational cost for each example. At the same time, the larger pool of experts allows the model to capture more subtle variations in the data, such as differences in pose, lighting, and demographic diversity present in the WebFace12M dataset. As a result, FaceMoE learns a richer set of feature subspaces, which enhances its robustness on both standard and challenging benchmarks, even after fine-tuning on downstream datasets.

Moreover, sparse routing serves as an implicit regularizer. FaceMoE updates only a fraction of the model parameters in each mini-batch, which helps reduce co-adaptation among experts and protects against overfitting, even as the dataset continues to grow. This built-in regularization becomes increasingly valuable when training on tens of millions of images, as it ensures that each expert develops a distinct specialization rather than converging into redundant representations. In addition, the computational efficiency of mixture-of-experts models allows for high model capacity while keeping the floating point operations per example manageable. This efficiency enables longer and more thorough training within a fixed compute budget, allowing FaceMoE to fully leverage the extensive data available in WebFace12M. Together, these factors explain why increasing the size of the pre-training dataset leads to consistent and cost-effective improvements in FaceMoE's recognition performance during both pre-training and downstream fine-tuning.

### B.10 PERFORMANCE UNDER SYNTHETIC DEGRADATIONS

| Method | LFW | CFP-FF | AgeDB-30 | Expert 0 | Expert 1 | Expert 2 |
|---|---|---|---|---|---|---|
| FaceMoE | 99.75 | 99.86 | 97.45 | 33.4 | 33.6 | 33.0 |
| Gaussian std = 1 | 99.71 | 99.81 | 97.30 | 33.2 | 35.2 | 31.6 |
| Gaussian std = 5 | 76.53 | 69.77 | 55.90 | 32.7 | 37.0 | 30.3 |
| JPEG 30% | **99.6** | **99.65** | **96.93** | 33.5 | 34.6 | 31.9 |

Table B.10: Performance under synthetic degradations

We perform a stress test on FaceMoE under synthetic degradations. We synthetically apply Gaussian blur and JPEG compression to the probe images while keeping the gallery fixed, and we report the verification accuracy on LFW/CFP-FP/AgeDB-30 and retrieval performance on TinyFace for occlusion robustness. As shown in Table B.10, mild degradations (Gaussian $\sigma = 1$, JPEG 30%) lead to only marginal changes, indicating that the routing mechanism remains stable and continues to select suitable experts even when image quality is moderately reduced. Under severe blur ($\sigma = 5$), performance drops substantially particularly on AgeDB-30 consistent with the fact that heavy low-pass filtering removes discriminative identity cues that no expert can fully compensate for. The routing statistics adapt and show increased activation of experts ($\approx 37\%$) specialized for low-frequency representations, confirming the hypothesized adaptive behavior. For occlusion, we evaluate on the TinyFace benchmark, which includes masks over the eyes, mouth, and nose. These landmark occlusions severely reduce the available identity information, as they block key facial regions used for recognition, which in turn leads to a significant drop in absolute performance. Although absolute performance is lower due to the extreme occlusions, the model maintains stable rank-1/5/20 trends.

These results demonstrate that FaceMoE adapts its routing under synthetic degradations, and performance only degrades significantly when identity information becomes intrinsically unrecoverable.

### B.11 INFERENCE COMPUTATION ANALYSIS

We perform an inference computation analysis and measure inference latency, peak memory consumption, and throughput on two GPU configurations: an NVIDIA RTX A5000 (representing a resource-constrained setting) and an NVIDIA A6000. All experiments were conducted on the same dataset (161,599 images) with a batch size of 800, and include the full routing overhead of our

| Metric | RTX A5000 | RTX A6000 |
|---|---|---|
| Total Images | 161,599 | 161,599 |
| Batch Size | 800 | 800 |
| Average Batch Latency (ms) | 8109.92 | 6170.72 |
| Per-Image Latency (ms) | 10.27 | 7.80 |
| Throughput (fps) | 97.32 | 128.19 |
| Peak GPU Memory Usage (GB) | 10.40 | 10.40 |

Table B.11: Inference latency, throughput, and memory metrics (including router overhead) on RTX A5000 and RTX A6000.

method. As shown in the Table B.11, the A6000 provides a substantial improvement in throughput (128.19 fps vs. 97.32 fps) and reduced average batch latency, while peak GPU memory usage remains identical across GPUs (10.40 GB). These results demonstrate that our method scales efficiently across hardware tiers and maintains practical latency/memory characteristics even on a constrained GPU such as the A5000.

## B.12 QUANTITATIVE EVIDENCE OF SELECTIVE DRIFT

| Layer | CKA Similarity |
|---|---|
| patch_embed | 0.999867 |
| layers.0.blocks.0 | 0.998419 |
| layers.0.blocks.1 | 0.996523 |
| layers.1.blocks.0 | 0.995274 |
| layers.1.blocks.1 | 0.995154 |
| layers.1.blocks.2 | 0.994973 |
| layers.1.blocks.3 | 0.995113 |
| layers.1.blocks.4 | 0.994977 |
| layers.1.blocks.5 | 0.995119 |
| layers.1.blocks.6 | 0.995238 |
| layers.1.blocks.7 | 0.995003 |
| layers.1.blocks.8 | 0.994763 |
| layers.1.blocks.9 | 0.995177 |
| layers.1.blocks.10 | 0.994537 |
| layers.1.blocks.11 | 0.994277 |
| layers.1.blocks.12 | 0.993579 |
| layers.1.blocks.13 | 0.993211 |
| layers.1.blocks.14 | 0.991946 |
| layers.1.blocks.15 | 0.990840 |
| layers.1.blocks.16 | 0.989383 |
| layers.1.blocks.17 | 0.983919 |
| layers.2.blocks.0 | 0.977410 |
| layers.2.blocks.1 | 0.808624 |
| norm | 0.877010 |
| feature_layer | 0.867713 |

Table B.12: CKA similarity for each layer

In this subsection, we provide quantitative evidence of reduced forgetting and selective drift which makes our paper stronger. To address this, we performed additional analyses comparing the HR-pretrained model with the LR-finetuned FaceMoE model.

(1) CKA-based representational drift. We compute CKA similarity layer-by-layer to measure representational changes. As shown in Table B.12, most layers maintain extremely high similarity (0.99+), including patch embedding and early/mid transformer blocks, indicating negligible forgetting. Drift gradually increases only in deeper layers (e.g., layers.1.blocks.17: 0.984; layers.2.blocks.0: 0.977),

| Layer | CKA Similarity |
|---|---|
| layers.2.blocks.1 | 0.808624 |
| feature_layer | 0.867713 |
| norm | 0.877010 |
| layers.2.blocks.0 | 0.977410 |
| layers.1.blocks.17 | 0.983919 |

Table B.13: Top 5 Most Changed Layers (Lowest CKA).

| Expert | L2 Shift (Magnitude) | Expert Shift (%) |
|---|---|---|
| Expert 0 | 45.9942 | 7.9835% |
| Expert 1 | 46.0058 | 8.0334% |
| Expert 2 | 45.6348 | 8.0775% |

Table B.14: L2 shift and approximate expert shift percentage.

with the largest adaptation occurring in layers.2.blocks.1 (0.8086) and the final feature layer (0.8677). These are precisely the layers responsible for high-level identity semantics, supporting our claim that FaceMoE preserves foundational HR features while adapting selectively for LR data.

(2) Localizing drift (lowest-CKA layers). The five most changed layers (Table B.13) are exclusively in the deepest stage and output head. This indicates targeted high-level adaptation rather than global drift, supporting our claim that forgetting is minimized and adaptation is concentrated on identity-semantic layers.

(3) Expert parameter update We also measure the L2 parameter shift for each expert. As reported in Table B.14, all experts undergo only a small relative shift of 8%, despite being trainable during LR finetuning. This modest drift indicates that FaceMoE adapts sufficiently to the LR domain while preserving the majority of the HR-pretrained structure. The small magnitude of change across experts quantitatively supports our claim that MoE reduces forgetting by enabling controlled, localized adaptation rather than wholesale parameter updates.

## B.13 HYPERPARAMETER SENSITIVITY

| $\lambda$ | TinyFace | | | BRIAR | | |
|---|---|---|---|---|---|---|
| | Rank-1 | Rank-5 | Rank-20 | 0.01% | 0.10% | 1% |
| 1 | 76.09 | 79.82 | 81.66 | 42.27 | 61.53 | 81.22 |
| 5 | 76.48 | 0.06 | 0.06 | 42.56 | 61.61 | 81.52 |
| 9 | 76.31 | 79.83 | 82.24 | 42.30 | 61.68 | 81.43 |
| 10 | 76.18 | 79.69 | 81.75 | 42.36 | 61.47 | 81.27 |
| 11 | 76.42 | 79.90 | 82.00 | 42.56 | 61.52 | 81.62 |
| 15 | 76.18 | 79.77 | 82.10 | 42.46 | 61.38 | 81.21 |
| 100 | 76.31 | 79.66 | 82.05 | 42.34 | 61.52 | 81.41 |

Table B.15: Performance metrics for TinyFace and BRIAR across different $\lambda$ values.

We perform a sensitivity analysis for the loss-weighting hyperparameter $\lambda$, which jointly scales the router $z$-loss and load-balancing loss in the total objective:

$$L_{\text{total}} = L_{\text{face}} + \lambda \left( L_z + L_{\text{balance}} \right).$$

The table above reports performance obtained by sweeping

$$\lambda \in \{1, 5, 9, 10, 11, 15, 100\}$$

on two datasets (TinyFace and BRIAR):

- **TinyFace:** Rank-1 accuracy ranges from 76.09% to 76.48% (a spread $< 0.4$ percentage points). Rank-5 and Rank-20 accuracies vary by only about $0.3$ percentage points across the entire sweep.
- **BRIAR (Protocol 3.1):** TAR@FAR=0.01% ranges from 42.27% to 42.56%, TAR@0.10% from 61.38% to 61.68%, and TAR@1% from 81.21% to 81.62%. All metrics vary by roughly $0.4$ percentage points or less.

These results show that performance is *not* brittle with respect to $\lambda$, even when varied over more than two orders of magnitude. In particular, there is a clear performance plateau for

$$\lambda \in [5, 15],$$

within which both TinyFace and BRIAR metrics remain effectively unchanged.

## B.14 ROUTING STABILITY AND CAUSALITY

| Attack | Rank-1 | Rank-5 | Rank-20 | Expert 0 | Expert 1 | Expert 2 |
|--------|--------|--------|---------|----------|----------|----------|
| FGSM | 74.88 | 79.09 | 81.63 | 33.2 | 33.5 | 33.3 |
| PGD | 73.41 | 77.54 | 80.06 | 32.6 | 34.6 | 32.8 |
| MIM | 74.14 | 78.32 | 80.87 | 33.1 | 32.7 | 34.2 |

Table B.16: Performance comparison across attacks and experts.

| | TinyFace | | |
|---|---|---|---|
| | Rank-1 | Rank-5 | Rank-20 |
| FaceMoE | 76.18 | 79.69 | 81.75 |
| Switch Expert | 69.68 | 75.40 | 79.07 |
| Drop Expert | | | |
| 0 | 75.05 | 78.99 | 81.65 |
| 1 | 75.10 | 78.88 | 81.62 |
| 2 | 75.08 | 78.91 | 81.94 |
| 0, 1 | 69.68 | 74.83 | 78.75 |
| 1, 2 | 69.98 | 75.80 | 80.12 |
| 0, 2 | 68.56 | 74.38 | 78.13 |

Table B.17: Performance on TinyFace when dropping individual or pairs of experts, and switching experts.

We conducted experiments to show the performance across perturbations, and further performed experiments by dropping and switching experts, to strengthen our claim that the performance is achieved by the proposed design and not by increased capacity.

**1. Routing is Stable Across Perturbations** To assess stability, we measure token-to-expert assignments under several common perturbations (FGSM, PGD, MIM). As shown in Table B.16, the routing distribution across the three experts remains highly stable:

- Under all perturbations, expert usage stays nearly uniform ($\approx 33\%$ per expert), with $< 2\%$ deviation across attacks.
- Even stronger iterative attacks (PGD, MIM) do not cause expert collapse or oscillation. This consistency shows that the router is not sensitive to small input perturbations such as adversarial noise, and that token assignments converge to stable semantic regions, not noise-driven fluctuations. Therefore, routing behavior is robust and not an unstable byproduct of MoE capacity.

**2. Controlled Expert Ablations Reveal Causal Contribution of Specialization** To evaluate whether FaceMoE's performance stems from learned specialization rather than increased parameters, we run two sets of interventions as shown in Table B.17:

*(a) Dropping Individual Experts*

Removing any one expert while keeping model capacity nearly unchanged produces only a small drop ($\sim 1.0\%$) in Rank-1 relative to full FaceMoE:

- Expert 0 dropped: 75.05
- Expert 1 dropped: 75.10
- Expert 2 dropped: 75.08
- Full model: 76.18

This small but consistent degradation indicates that each expert contributes complementary information rather than redundant capacity. Since our router uses top-$k$ routing with $k = 2$, every token is always processed by two experts, ensuring that even after removing one expert, at least one of the originally assigned specialists is still active. This limits the performance drop while still revealing the non-redundant contribution of each expert.

*(b) Dropping Pairs of Experts (forcing single-expert routing)*

When two experts are removed, routing collapses into a single FFN branch. This mirrors a standard transformer's feed-forward layer capacity but accuracy drops drastically:

- Experts {0,1}: 69.68
- Experts {1,2}: 69.98
- Experts {0,2}: 68.56

The $\sim 6 - 8\%$ absolute drop demonstrates that increased capacity alone cannot account for performance gains. If raw capacity were the cause, single-expert models (same depth, same FLOPs) would not collapse this sharply. Instead, this strongly supports specialization as the mechanism driving improvements.

Together, these interventions demonstrate a causal chain:

- Routing remains stable across perturbations (Table B.16).
- Experts are not interchangeable (Switch Expert $\rightarrow$ significant drop).
- Experts are not redundant (dropping experts reduces performance).

Thus, improvements are not attributable to mere extra parameters but arise from structured expert specialization and resolution-aware routing, as intended in the design.

## C  ADDITIONAL IMPLEMENTATION DETAILS

These are the additional details provided in addition to the ones mentioned in the main paper. Our base architecture for all experiments is the Swin-B (Swin Transformer - Base), which serves as the backbone for the FaceMoE model. To provide a rough estimate of computational requirements, we report training times for various configurations of the number of experts ($N$) and the number of active experts per token ($k$). These estimates are not intended for comparison, as the experiments were conducted on both NVIDIA A6000 (48GB) and A5000 (24GB) GPUs, leading to variability in runtime. Specifically, training times (in hours) are approximately: 49 for (N=2, k=1), 57 for (N=3, k=1), 81 for (N=3, k=2), 120 for (N=3, k=3), 49 for (N=4, k=1), 50 for (N=4, k=2), and 88 for (N=4, k=3). To ensure a consistent and fair evaluation, we retrained the CosFace, ArcFace, and AdaFace baselines. For other baselines, we report results as presented in their respective original publications. All models and experiments are implemented in PyTorch and run across eight GPUs.

## D  EXPERT ACTIVATION MAPS

To gain insight into how each expert specializes before and after TinyFace finetuning, we visualize their spatial activation patterns on a few facial images, as shown in Figure 6. Each row presents the activations of all $k$ experts for a single input image.

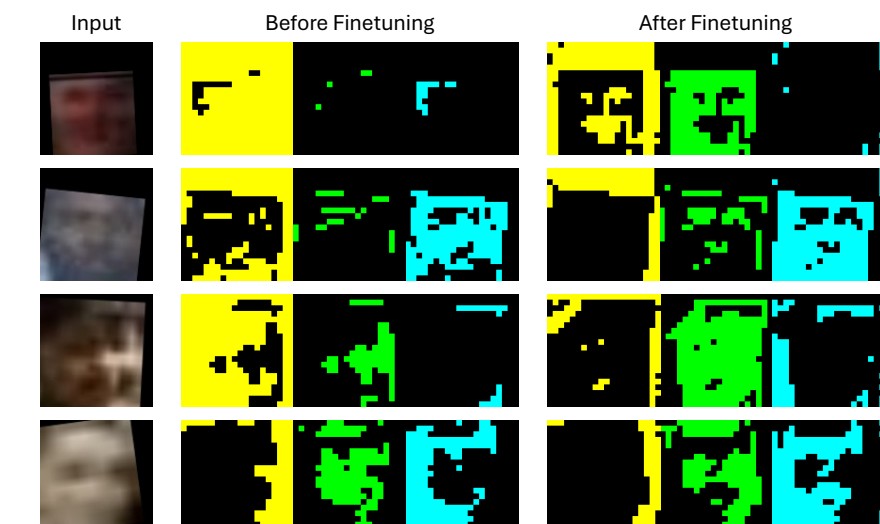

Figure 6: **Expert activation maps before and after TinyFace finetuning.** Each row shows the spatial activations of all $k$ experts on a input face image, before and after TinyFace finetuning. (Left) After pretraining on WebFace4M, experts exhibit broadly overlapping activations focusing on general facial regions (eyes, nose bridge, mouth outline). (Right) Following TinyFace finetuning, experts specialize on distinct, localized cues (eye corners, nose shape, cheek textures, etc.), yielding complementary attention patterns better suited to low-resolution face recognition.

**Pretraining on WebFace4M:** Before undergoing any adaptation to the TinyFace dataset, the model is pretrained for face recognition using the large-scale WebFace4M dataset. During this phase, all experts learn from a diverse collection of face images that vary in quality and pose, ranging from frontal to non-frontal views. As a result, their activation maps tend to highlight broad, coarse-grained regions, such as the overall outline of the face, the contours of the eyes, and the mouth area. There is substantial overlap between the activation patterns of different experts, suggesting that in the absence of further specialization, the experts tend to redundantly focus on the most generally discriminative facial features, such as the eyes and the bridge of the nose. These features remain consistently informative across a wide range of identities and imaging conditions.

**After TinyFace Finetuning:** Following finetuning on the TinyFace dataset, which consists of low-resolution face crops extracted from unconstrained scenes, the experts begin to capture more localized and complementary features. The activation maps demonstrate that individual experts now respond to specific subregions or patterns. Some experts focus closely on areas such as the eye corners and eyelid textures, which are particularly important in low-resolution scenarios. Others concentrate on features such as the shape of the nose or the contours of the mouth. Additional experts respond to compound patterns, including shadows on the cheeks or the silhouettes of ears. This diversity in focus reflects the model's adaptation to the characteristics of the TinyFace dataset. By distributing representational capacity across multiple experts, the network learns that fine-grained, region-specific textural cues are essential for distinguishing identities when the global structural features of the face are degraded due to low resolution.

The transition from broadly overlapping activations in the WebFace4M pretraining phase to highly specialized and non-redundant activation maps after TinyFace finetuning highlights the effectiveness of the MoE architecture for domain adaptation. In low-resolution settings, relying on a single shared backbone imposes a trade-off between capturing global structures and preserving fine-grained local details. In contrast, the MoE framework enables different sub-networks to allocate their representational capacity to the most reliable cues for the target domain. First, the model demonstrates robustness to resolution degradation. Experts that are tuned to textural patterns, such as the micro-structure of skin around the eyes, retain their discriminative ability even when the overall facial shape becomes indistinct. Second, the architecture facilitates the integration of complementary evidence. By aggregating signals from multiple specialized experts, the model can combine weak, localized features into a coherent and robust identity representation. Finally, the approach allows for efficient

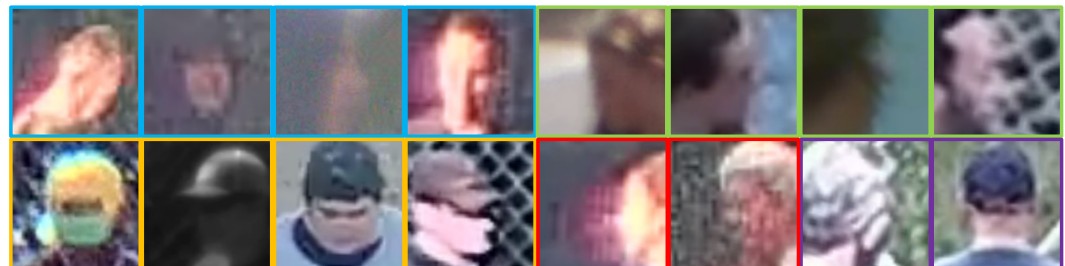

Figure 7: Failure Case Analysis of FaceMoE model on the BRIAR dataset.

adaptation. Only a subset of experts needs to specialize deeply in the new domain, while others can maintain their generalist knowledge from pretraining. This division of labor ensures a balanced trade-off between plasticity and stability.

***These activation patterns offer clear evidence that finetuning on low-resolution dataset induces functional specialization among experts, enabling the model to perform effectively in challenging, low-resolution face recognition tasks.***

## E   FAILURE CASE ANALYSIS

To diagnose the remaining weaknesses of our FaceMoE model, we conducted a detailed examination of representative failure cases on the BRIAR probe set as shown in Figure 7. We identified five dominant scenarios that consistently lead to recognition errors. First, **extremely low-resolution** face crops, typically below approximately $8 \times 8$ pixels, contain too little texture or shape information for reliable matching. This causes the expert ensemble's activations to become noisy and prone to errors. Second, **extreme head poses**, such as profiles or tilts greater than $60$ degrees, often result in facial landmarks moving outside the visible region. In these situations, experts trained on frontal-view patterns perform poorly. Third, **heavy occlusion** caused by items like masks, caps, or scarves can obscure important facial regions. As a result, the experts struggle to extract meaningful unoccluded features, which increases confusion with other identities. Fourth, **atmospheric turbulence**, including visual distortions such as heat shimmer and motion blur that are common in long-range surveillance, disrupts the spatial consistency of facial features. These effects fragment the activation maps and reduce the model's ability to form coherent representations. Finally, **non-frontal views**, where subjects never present a clear frontal face during a sequence, prevent the model from obtaining a stable canonical reference. Consequently, even viewpoint-specialized experts are unable to generate consistent embeddings, leading to recognition failures. These failure modes illustrate that, while FaceMoE is effective in handling low-resolution images, it remains vulnerable to conditions that obscure or dynamically distort facial information.

To evaluate the routing mechanism under extreme degradation, we visualize the activation maps for each expert in Figure 8. The figure contrasts success and failure cases by illustrating how activation patterns behave under challenging visual conditions. In successful examples, despite blur or moderate pose variations, the experts activate coherently around semantically meaningful facial regions, such as landmarks, contours, and stable low-frequency structures, allowing the network to extract sufficient identity cues. In failure cases, however, extreme pose, over/under-exposure, or severe occlusion disrupt this specialization: activation maps become diffuse, fragmented, or erroneously concentrated in non-informative regions. As shown in the failed inputs, experts often shift their focus to background patches or large smooth areas lacking discriminative detail, indicating that the model can no longer reliably localize or route tokens to the appropriate experts. This divergence between structured and unstable activation patterns highlights the sensitivity of low-resolution recognition models to severe degradations and explains why extreme angles, overexposure, and occlusion frequently lead to identity misclassification and degraded recognition performance.

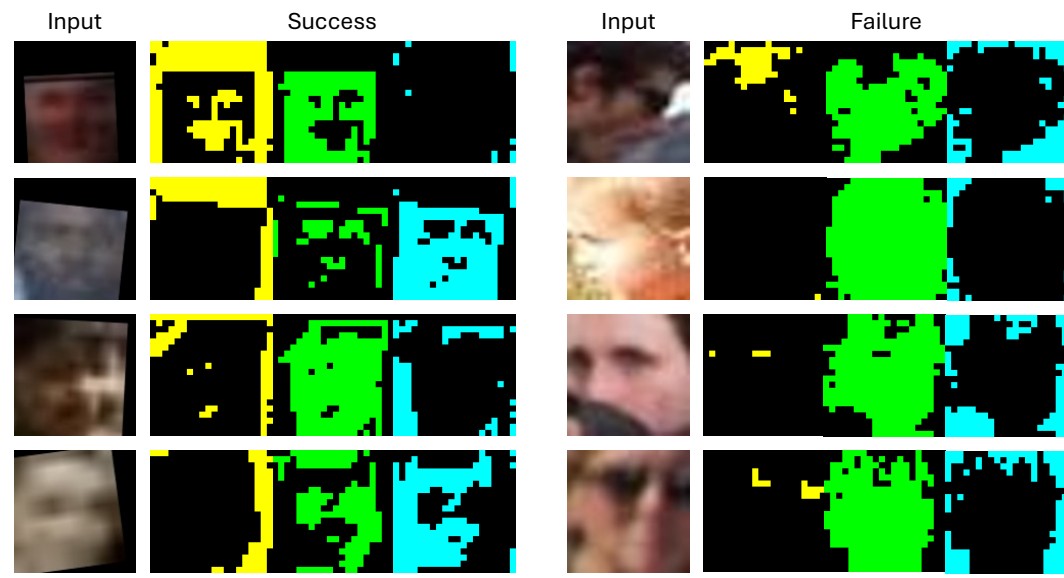

Figure 8: Comparison of activation maps for success and failure cases.

## F    Limitations and Future Work

Our training data, WebFace4M Zhu et al. (2021), is predominantly composed of Western, young, and light-skinned subjects. We have not yet incorporated balanced sampling, debiasing loss functions, or demographic-specific experts, which means the model may amplify existing biases. While Mixture-of-Experts (MoE) architectures are typically used to scale model capacity efficiently, their application in face recognition introduces unique challenges. We observe that increasing the number of experts ($N$) can lead to over-fragmentation and routing instability, which may negatively affect performance. Addressing these issues remains an important area for future work.

## G    Social Impact Statement

The proposed work, FaceMoE, presents a transformer-based Mixture of Experts (MoE) architecture that significantly advances low-resolution face recognition (LR-FR). FaceMoE enhances recognition performance on degraded or surveillance-quality imagery, offering the potential to improve operational effectiveness in domains such as public safety, disaster response, border control, and missing persons investigations. These improvements enable faster and more accurate identification in scenarios where traditional face recognition systems often underperform, particularly in time-sensitive or resource-constrained environments.

Beyond technical improvements, the broader societal implications of these advancements merit careful consideration. As face recognition systems become increasingly capable of identifying individuals from poor-quality images, their deployment in everyday settings such as public transit, city surveillance, or consumer electronics is likely to accelerate. This trend could contribute to a societal shift in which continuous identity tracking becomes normalized, potentially eroding expectations of anonymity and reshaping perceptions of privacy in public spaces. The widespread presence of such systems may also influence individual behavior and social engagement, particularly in communities that are already subject to heightened surveillance.

Furthermore, access to advanced recognition systems like FaceMoE may not be distributed evenly. Organizations with greater financial and technical resources are more likely to benefit from such technologies, which could deepen existing disparities in areas such as law enforcement, national security, and institutional capacity. Public trust in face recognition systems depends not only on their technical performance but also on how transparently and equitably they are implemented. To

ensure that FaceMoE contributes positively to society, its deployment in real-world applications must be supported by inclusive access, meaningful public dialogue, and policies that emphasize fairness, accountability, and the protection of civil liberties.

