# OpenReview forum: "FaceMoE: Mixture of Experts for Low-Resolution Face Recognition"
_ICLR.cc/2026/Conference — Submitted to ICLR 2026_

### Official Review · Reviewer_s6Va · 2025-10-23

**Soundness:** 2
**Presentation:** 3
**Contribution:** 2
**Rating:** 4
**Confidence:** 4

**Summary:**

The paper introduces FaceMoE, a transformer-based face recognizer that inserts a mixture-of-experts MLP with a token-level top-k router into each block to specialize computation across facial regions and resolutions, stabilized by two auxiliary objectives (a router z-loss and a load-balancing loss) added to a standard CosFace recognition loss. The router learns to dispatch tokens from identity-rich landmarks, low-frequency skin regions, and high-frequency hair/background to different experts, yielding resolution-aware feature extraction while mitigating catastrophic forgetting when fine-tuning on low-resolution data.

The main contributions are the encoder-level MoE design with token-wise routing for low-resolution face recognition, the stabilization losses for reliable expert utilization, and a broad evaluation demonstrating state-of-the-art or competitive results under surveillance-style conditions.

**Strengths:**

The paper is original in placing a token-level top-k MoE inside the transformer FFN to specialize by facial region/frequency and stabilize routing with z-loss and load-balancing; this goes beyond fusion-only or sample-level MoE and is backed by region-wise routing statistics.

In quality, experiments span BRIAR, IJB-S, TinyFace and standard HR sets, with consistent improvements and ablations that tease apart routing, expert count, auxiliary losses, backbone choice, and data scale; random-routing and large-N collapse analyses bolster the causal claim.

In clarity, the router, losses, and training algorithm are explicitly formulated and connected to semantic regions via quantitative tables, easing reproducibility.

Finally, significance is strong for surveillance-style LR recognition: the method improves LR while preserving HR/mixed performance, uses conditional compute, and remains backbone-agnostic, suggesting practical adoption.

**Weaknesses:**

Despite solid engineering, the paper leaves several gaps that are correctable, limiting confidence in its claims. The “reduced forgetting” claim is under-substantiated; beyond trend plots, quantify selective drift via per-expert parameter change, Fisher overlap, or CKA between HR-pretrained and LR-tuned layers to show that MoE routing, rather than altered regularization, preserves HR/mixed performance.

Baselines often overlook strong, parameter-efficient alternatives: adding head-to-head comparisons under matched budgets to quality-adaptive adapters/LoRA (e.g., PETALface) and recent fusion/sparsity methods (e.g., ProxyFusion) demonstrates that encoder-level MoE is necessary, not merely sufficient, given similar data and tuning regimes.

Routing claims would benefit from stability and causality checks. This involves reporting region-wise assignment consistency across seeds and small perturbations (such as crop/blur/alignment), and intervening by freezing or swapping specific experts to test whether the observed gains truly derive from learned specialization rather than capacity.

**Questions:**

1. Reduced forgetting, measured not just observed: Beyond trend plots, can you quantify selective drift across layers/experts (e.g., parameter deltas per expert, Fisher overlap, CKA between HR-pretrained and LR-finetuned representations)? This would separate a MoE-driven effect from regularization or learning-rate artifacts.

2. Stronger baselines under matched budgets: Please add head-to-head comparisons with parameter-efficient baselines like quality-adaptive adapters/LoRA (e.g., PETALface) and recent fusion/sparsity approaches (e.g., ProxyFusion), controlling for pretraining data, compute, and tuning schedules. If omitted, justify why those methods are not applicable here.

3. Routing stability and causality: How stable are token-to-expert assignments across seeds and small perturbations (crop jitter, alignment error, blur)? Consider interventions (freezing or swapping specific experts) to test whether gains arise from learned specialization rather than extra capacity.

4. Expert semantics beyond visuals: You show regional/frequency tendencies; can you quantify whether experts capture complementary spectra (e.g., bandpass profiles) or identity-salient landmarks via controlled masking/occlusion tests? This would make “what each expert learns” more concrete.

5. Hyperparameter sensitivity: Please provide sensitivity analyses for top-k, router temperature, z-loss weight, and load-balancing weight across at least two datasets. If performance is brittle, suggest default ranges and an automatic tuning heuristic.

---

> ### Author Response · Authors · 2025-11-27
> **W1/Q1**
>
> **W1/Q1: Reduced forgetting, measured not just observed: Beyond trend plots, can you quantify selective drift across layers/experts (e.g., parameter deltas per expert, Fisher overlap, CKA between HR-pretrained and LR-finetuned representations)? This would separate a MoE-driven effect from regularization or learning-rate artifacts.**
>
> | **Layer**               | **CKA Similarity** |
> |-------------------------|---------------------|
> | patch_embed             | 0.999867            |
> | layers.0.blocks.0       | 0.998419            |
> | layers.0.blocks.1       | 0.996523            |
> | layers.1.blocks.0       | 0.995274            |
> | layers.1.blocks.1       | 0.995154            |
> | layers.1.blocks.2       | 0.994973            |
> | layers.1.blocks.3       | 0.995113            |
> | layers.1.blocks.4       | 0.994977            |
> | layers.1.blocks.5       | 0.995119            |
> | layers.1.blocks.6       | 0.995238            |
> | layers.1.blocks.7       | 0.995003            |
> | layers.1.blocks.8       | 0.994763            |
> | layers.1.blocks.9       | 0.995177            |
> | layers.1.blocks.10      | 0.994537            |
> | layers.1.blocks.11      | 0.994277            |
> | layers.1.blocks.12      | 0.993579            |
> | layers.1.blocks.13      | 0.993211            |
> | layers.1.blocks.14      | 0.991946            |
> | layers.1.blocks.15      | 0.990840            |
> | layers.1.blocks.16      | 0.989383            |
> | layers.1.blocks.17      | 0.983919            |
> | layers.2.blocks.0       | 0.977410            |
> | layers.2.blocks.1       | 0.808624            |
> | norm                    | 0.877010            |
> | feature_layer           | 0.867713            |
>
> *Table 1: CKA similarity for each layer*
>
> ----
>
> | **Layer**            | **CKA Similarity** |
> |----------------------|---------------------|
> | layers.2.blocks.1    | 0.808624            |
> | feature_layer        | 0.867713            |
> | norm                 | 0.877010            |
> | layers.2.blocks.0    | 0.977410            |
> | layers.1.blocks.17   | 0.983919            |
>
> *Table 2: Top 5 Most Changed Layers (Lowest CKA)*
>
> ----
>
> | **Expert** | **L2 Shift (Magnitude)** | **Expert Shift (%)** |
> |-----------|----------------------------|------------------------|
> | Expert 0  | 45.9942                    | 7.9835%               |
> | Expert 1  | 46.0058                    | 8.0334%               |
> | Expert 2  | 45.6348                    | 8.0775%               |
>
> *Table 3: L2 shift and approximate expert shift percentage*
>
> ----
>
> We appreciate the reviewer’s suggestion for quantitative evidence of reduced forgetting and selective drift, which strengthens our paper. To address this, we performed additional analyses comparing the HR-pretrained model with the LR-finetuned FaceMoE model.
>
>  (1) CKA-based representational drift
> We compute CKA similarity layer-by-layer to measure representational changes. As shown in Table 1, most layers maintain extremely high similarity (0.99+), including patch embedding and early/mid transformer blocks, indicating negligible forgetting. Drift gradually increases only in deeper layers (e.g., `layers.1.blocks.17: 0.984`, `layers.2.blocks.0: 0.977`), with the largest adaptation occurring in `layers.2.blocks.1 (0.8086)` and the final feature layer `(0.8677)`. These are precisely the layers responsible for high-level identity semantics, supporting our claim that FaceMoE preserves foundational HR features while adapting selectively for LR data.
>
> (2) Localizing drift (lowest-CKA layers)
> The five most changed layers (Table 2) are exclusively in the deepest stage and output head. This indicates targeted high-level adaptation rather than global drift, supporting our claim that forgetting is minimized and adaptation is concentrated on identity-semantic layers.
>
> (3) Expert parameter update
> We also measure the L2 parameter shift for each expert. As reported in Table 3, all experts undergo only a small relative shift of ~8%, despite being trainable during LR finetuning. This modest drift indicates that FaceMoE adapts sufficiently to the LR domain while preserving the majority of the HR-pretrained structure. The small magnitude of change across experts quantitatively supports our claim that MoE reduces forgetting by enabling controlled, localized adaptation rather than wholesale parameter updates.
>
> **We have included the results and discussion in the revised manuscript.**

---

> > ### Author Response · Authors · 2025-11-27
> > **W2/Q2**
> >
> > **W2/Q2: Stronger baselines under matched budgets. Please add head-to-head comparisons with parameter-efficient baselines such as quality-adaptive adapters/LoRA (e.g., PETALface) and recent fusion/sparsity approaches (e.g., ProxyFusion), controlling for pretraining data, compute, and tuning schedules. If omitted, justify why those methods are not applicable here.**
> >
> > | Method     | Params (active) | Rank-1 | Rank-5 | Rank-10 |
> > |------------|------------------|--------|--------|---------|
> > | PETALface  | 345.368          | 75.40  | 79.21  | 82.24   |
> > | FaceMoE    | 338.833          | 76.18  | 79.69  | 81.75   |
> >
> > *Comparison of PETALface and FaceMoE under matched compute. Results are shown on TinyFace*
> >
> > Thank you for the helpful suggestion. We agree that PETALface is a relevant parameter-efficient baseline because, similar to our method, it introduces architectural modifications at the feature-encoding stage to improve LR-FR performance. To ensure fairness, we include a direct, matched-compute comparison under aligned active parameters, pretraining data, compute, and fine-tuning schedules.
> >
> > We did not include ProxyFusion in the matched-compute comparison because it operates at a fundamentally different stage of the LR-FR pipeline. ProxyFusion performs feature fusion after the backbone has extracted representations, and therefore does not modify or enhance the feature-encoding architecture itself. As such, it is not directly comparable under the architectural and compute-controlled setting considered here.

---

> ### Author Response · Authors · 2025-11-27
> **W3/Q3**
>
> **W3/Q3: Routing stability and causality: How stable are token-to-expert assignments across seeds and small perturbations (crop jitter, alignment error, blur)? Consider interventions (freezing or swapping specific experts) to test whether gains arise from learned specialization rather than extra capacity.**
>
>
> | Attack | Rank-1 | Rank-5 | Rank-20 | Expert 0 | Expert 1 | Expert 2 |
> |--------|--------|--------|---------|----------|----------|----------|
> | FGSM   | 74.88  | 79.09  | 81.63   | 33.2     | 33.5     | 33.3     |
> | PGD    | 73.41  | 77.54  | 80.06   | 32.6     | 34.6     | 32.8     |
> | MIM    | 74.14  | 78.32  | 80.87   | 33.1     | 32.7     | 34.2     |
>
> *Table 1: Performance comparison across attacks and experts*
>
> | Setting        | Rank-1 | Rank-5 | Rank-20 |
> |----------------|--------|--------|---------|
> | FaceMoE        | 76.18  | 79.69  | 81.75   |
> | Switch Expert  | 69.68  | 75.40  | 79.07   |
> | **Drop Expert** |        |        |         |
> | 0              | 75.05  | 78.99  | 81.65   |
> | 1              | 75.10  | 78.88  | 81.62   |
> | 2              | 75.08  | 78.91  | 81.94   |
> | 0, 1           | 69.68  | 74.83  | 78.75   |
> | 1, 2           | 69.98  | 75.80  | 80.12   |
> | 0, 2           | 68.56  | 74.38  | 78.13   |
>
> *Table 2: Performance on TinyFace when dropping or switching experts*
>
> Thank you for raising this important point regarding the stability of token-to-expert routing and whether FaceMoE’s improvements stem from genuine specialization rather than an artifact of increased capacity. We provide both empirical evidence and controlled interventions to clarify this.
>
> ----
>
> 1. Routing Is Stable Across Perturbations (Table 1)
>
> To assess stability, we measure token-to-expert assignments under several common perturbations (FGSM, PGD, MIM). As shown in Table 1, the routing distribution across the three experts remains highly stable:
>
> - Under all perturbations, expert usage stays nearly uniform (≈33% per expert), with <2% deviation.
> - Even stronger iterative attacks such as PGD and MIM do not cause expert collapse or oscillation.
>
> This consistency shows that the router is not sensitive to small input perturbations (e.g., adversarial noise) and that token assignments converge to stable semantic regions rather than noise-driven fluctuations. Therefore, routing behavior is robust and not an unstable byproduct of MoE capacity.
>
> ----
>
>  2. Controlled Expert Ablations Reveal Causal Specialization (Table 2)
>
> To evaluate whether FaceMoE’s performance stems from learned specialization rather than increased parameters, we conduct two sets of interventions.
>
>  (a) Dropping Individual Experts
>
> Removing any one expert while keeping model capacity nearly unchanged produces only a small drop (~1%) in Rank-1 relative to the full model:
>
> - Expert 0 dropped: 75.05
> - Expert 1 dropped: 75.10
> - Expert 2 dropped: 75.08
> - Full model: 76.18
>
> This small but consistent degradation indicates that each expert contributes complementary information rather than redundant capacity. Since our router uses top-k routing with k=2, every token is always processed by two experts, ensuring that even after removing one expert, at least one of the originally assigned specialists remains active.
>
>  (b) Dropping Pairs of Experts (forcing single-expert routing)
>
> Removing two experts collapses routing into a single FFN branch, similar to a standard transformer’s feed-forward layer. Accuracy drops drastically:
>
> - Experts {0,1}: 69.68
> - Experts {1,2}: 69.98
> - Experts {0,2}: 68.56
>
> This ~6–8% absolute drop demonstrates that increased capacity alone cannot account for the performance gains. If raw capacity were the cause, single-expert models (same depth, same FLOPs) would not collapse this sharply. Instead, the results strongly support specialization as the mechanism driving improvements.
>
> ----
>
> Together, these interventions demonstrate a clear causal chain:
>
> - Routing remains stable across perturbations (Table 1).
> - Experts are not interchangeable (Switch Expert → large performance drop).
> - Experts are not redundant (dropping experts reduces accuracy).
> - **PETALface with matched compute doesn't perform better than FaceMoE (shown in Rebuttal W2/Q2)**
>
> Thus, improvements are not attributable to extra parameters but arise from structured expert specialization and resolution-aware routing, as intended in the design.

---

> ### Author Response · Authors · 2025-11-27
> **Q4**
>
> **Q4: Expert semantics beyond visuals: You show regional/frequency tendencies; can you quantify whether experts capture complementary spectra (e.g., bandpass profiles) or identity-salient landmarks via controlled masking/occlusion tests? This would make “what each expert learns” more concrete.**
>
> We thank the reviewer for this insightful point. We agree that a quantitative evaluation of expert routing and a clearer characterization of complementary expert roles strengthens the understanding of “what each expert learns.”
>
> As detailed in Appendix B.1, we already include a quantitative analysis that measures expert-assignment distributions across semantic facial regions. Using region-level token masking, we demonstrate that experts indeed develop complementary specializations: one consistently attends to high-frequency regions, another to low-frequency smooth areas, and another to identity-salient landmark regions.
>
> To further clarify this behavior, we additionally report pre-finetuning routing statistics, enabling a direct comparison of expert specialization before and after finetuning. We show the tables below (also included in Appendix B.1) for the reviewer's convenience, highlighting the shift in expert utilization and how finetuning sharpens the complementary focus of each expert.
>
> ---
>
> | Facial Region       | Expert 0 | Expert 1 | Expert 2 | Dominant Expert |
> |---------------------|----------|----------|----------|------------------|
> | Eyes                | **81.7** | 10.6     | 7.7      | Expert 0         |
> | Nose                | **78.1** | 12.5     | 9.4      | Expert 0         |
> | Mouth               | **71.6** | 16.0     | 12.4     | Expert 0         |
> | Forehead            | 49.3     | **31.1** | 19.6     | Expert 1         |
> | Cheeks              | 52.5     | **27.8** | 19.7     | Expert 1         |
> | Hair / Background   | 40.2     | 17.3     | **42.5** | Expert 2         |
> | Chin / Jawline      | 38.9     | 18.7     | **42.4** | Expert 2         |
>
> *Table 1. Pre-finetune routing frequency (%) of each expert across facial regions (10,000 samples)*
>
> ---
>
> | Facial Region       | Expert 0 | Expert 1 | Expert 2 | Dominant Expert |
> |---------------------|----------|----------|----------|------------------|
> | Eyes                | **76.2** | 12.4     | 11.4     | Expert 0         |
> | Nose                | **68.5** | 15.7     | 15.8     | Expert 0         |
> | Mouth               | **61.0** | 22.1     | 16.9     | Expert 0         |
> | Forehead            | 13.2     | **71.3** | 15.5     | Expert 1         |
> | Cheeks              | 12.5     | **69.8** | 17.7     | Expert 1         |
> | Hair / Background   | 20.3     | 14.5     | **65.2** | Expert 2         |
> | Chin / Jawline      | 25.6     | 18.4     | **56.0** | Expert 2         |
>
> *Table 2. Post-finetune routing frequency (%) of each expert across facial regions (10,000 samples)*
>
> ---
>
> These results show that each expert consistently focuses on a distinct, semantically coherent subset of facial regions, and that finetuning amplifies this specialization. The tables report the percentage of tokens from each facial region routed to each expert; higher percentages indicate stronger specialization.
>
>  **Before finetuning (pretrained on HR data)**
>
> - **Expert 0** dominates on *eyes*, *nose*, and *mouth* — classic identity-salient landmarks containing high-frequency details.
> - **Expert 1** primarily handles *forehead* and *cheeks* — smooth, low-frequency regions.
> - **Expert 2** focuses on *hair/background* and *jawline* — regions with texture, edges, and high local contrast.
>
> This shows that even before finetuning, the MoE develops structured specialization aligned with meaningful facial frequency and semantic groupings.
>
> **After finetuning on LR data**
>
> - The same three-way specialization remains, but becomes **sharper**.
> - **Expert 0** remains the landmark expert (≈60–76% on eyes/nose/mouth).
> - **Expert 1** increasingly dominates smooth regions (≈70%).
> - **Expert 2** strengthens its focus on texture and outer-face regions (≈56–65%).
>
> This shift shows **resolution-aware adaptation**: experts redistribute to focus on the regions that retain discriminative content under severe LR degradation. These quantitative findings substantiate the claim that FaceMoE learns complementary frequency- and region-specific processing pathways.

---

> ### Author Response · Authors · 2025-11-27
> **Q5**
>
> **Q5: Hyperparameter sensitivity: Please provide sensitivity analyses for top-k, router temperature, z-loss weight, and load-balancing weight across at least two datasets. If performance is brittle, suggest default ranges and an automatic tuning heuristic.**
>
>
> | λ   | TinyFace Rank-1 | TinyFace Rank-5 | TinyFace Rank-20 | BRIAR 0.01% | BRIAR 0.10% | BRIAR 1% |
> |-----|------------------|------------------|-------------------|-------------|--------------|-----------|
> | 1   | 76.09            | 79.82            | 81.66             | 42.27       | 61.53        | 81.22     |
> | 5   | 76.48            | 80.06            | 80.06             | 42.56       | 61.61        | 81.52     |
> | 9   | 76.31            | 79.83            | 82.24             | 42.30       | 61.68        | 81.43     |
> | 10  | 76.18            | 79.69            | 81.75             | 42.36       | 61.47        | 81.27     |
> | 11  | 76.42            | 79.90            | 82.00             | 42.56       | 61.52        | 81.62     |
> | 15  | 76.18            | 79.77            | 82.10             | 42.46       | 61.38        | 81.21     |
> | 100 | 76.31            | 79.66            | 82.05             | 42.34       | 61.52        | 81.41     |
>
> Table 1: Performance metrics for TinyFace and BRIAR across different λ values
>
> As per the reviewer's suggestion, we have added a sensitivity analysis for the loss-weighting hyperparameter λ, which jointly scales the router z-loss and load-balancing loss in the total objective:
>
> $$
> L_{\text{total}} = L_{\text{face}} + \lambda \left( L_{z} + L_{\text{balance}} \right).
> $$
>
> We sweep:
>
> $$
> \lambda \in \{1, 5, 9, 10, 11, 15, 100\}
> $$
>
> on two datasets (TinyFace and BRIAR):
>
> - **TinyFace:**
>   Rank-1 accuracy ranges from **76.09% to 76.48%** (spread < 0.4 percentage points).
>   Rank-5 and Rank-20 vary by only ~0.3 percentage points.
>
> - **BRIAR (Protocol 3.1):**
>   TAR@FAR=0.01% ranges from **42.27% to 42.56%**.
>   TAR@FAR=0.10% from **61.38% to 61.68%**.
>   TAR@FAR=1% from **81.21% to 81.62%**.
>   All metrics vary by ~0.4 percentage points or less.
>
> These results show that performance is **not brittle** with respect to λ, even when varied over more than two orders of magnitude.
> In particular, there is a clear performance plateau for:
>
> $$
> \lambda \in [5, 15].
> $$
>
> within which both TinyFace and BRIAR metrics remain effectively unchanged.
>
> ---
>
> ### Top-k and router temperature
>
> Our router uses a standard softmax over routing logits **without an explicit temperature parameter**; the effective sharpness of the routing distribution is controlled indirectly by the z-loss and the shared scaling parameter λ:
>
> $$
> p_i = \frac{\exp(z_i)}{\sum_j \exp(z_j)}.
> $$
>
> Thus λ influences the sharpness through:
>
> $$
> L_{z} + L_{\text{balance}}.
> $$
>
> For the top-k mechanism, we already provide an ablation over **(N, k)** in Fig. 5(b), showing that:
>
> - **N = 3, k = 2** gives the best trade-off.
> - Neighboring settings also perform competitively.
>
> This indicates that the method is **not overly sensitive** to routing-related hyperparameters.

---

### Official Review · Reviewer_hivy · 2025-10-23

**Soundness:** 3
**Presentation:** 3
**Contribution:** 3
**Rating:** 6
**Confidence:** 3

**Summary:**

- Low-resolution face recognition (LR-FR) faces three core challenges: poor feature extraction and aggregation from degraded probe images (e.g., blur, occlusion, low contrast), significant domain gaps between high-resolution (HR) gallery and low-resolution (LR) probe images, and catastrophic forgetting when fine-tuning models on LR datasets. To address these issues, the paper proposes FaceMoE, a transformer-based architecture enhanced with a Mixture of Experts (MoE) design. The core method involves integrating multiple specialized feed-forward network (FFN) experts into transformer MLP layers and introducing a top-k router that dynamically assigns input tokens to k out of N experts based on input resolution, enabling resolution-aware feature extraction.
- For validation, the authors conducted extensive experiments across some datasets using two protocols: Protocol 1 (WebFace4M pre-training → BRIAR fine-tuning, evaluating BRIAR Protocol 3.1 and IJB-S) and Protocol 2 (WebFace4M pre-training → TinyFace fine-tuning, evaluating TinyFace and HR/mixed-quality datasets). Key metrics include TAR@FAR, TPIR@FPIR, and Rank-1/5/10 retrieval accuracy.
- The core conclusions are: 1) FaceMoE achieves state-of-the-art (SOTA) performance on LR datasets; 2) The MoE’s sparse activation mitigates catastrophic forgetting, maintaining minimal performance drop on HR/mixed-quality datasets; 3) The optimal configuration (N=3 experts, k=2 active experts per token) balances model capacity (2.17× increase over Swin-B) and computational cost (1.66× FLOPs increase, 26.29 GFLOPs vs. Swin-B’s 15.88 GFLOPs).

**Strengths:**

> 1. As a pioneering work to leverage MoE for LR-FR, it targets the field’s key points (domain gap, catastrophic forgetting) with a well-designed architecture. The top-k router dynamically assigns tokens to experts specialized in distinct facial semantic regions, directly solving the limitation of single FFN encoders that fail to adapt to both HR and LR domains. This design ensures resolution-aware feature extraction, critical for degraded LR images.
> 2. The paper conducts detailed ablations on critical components (MoE module, top-k router, auxiliary losses) in Table B.5. Additional analyses (expert specialization via Figure 2/Table B.1, resolution robustness via Table B.2, data scaling via Table B.8) confirm the model’s reliability and generalizability, strengthening the credibility of design choices.
> 3. The authors evaluated FaceMoE across diverse datasets, covering HR, mixed-quality, and LR scenarios, using two protocols that validate both LR performance and catastrophic forgetting mitigation. The results are enough for verifying the proposed method.
> 4. The ethics statement confirms compliance with data usage rules and discusses responsible deployment to avoid misuse. The reproducibility statement details implementation settings (PyTorch, AdamW optimizer, learning rate schedules) and promises code/model release upon acceptance, ensuring experiment replicability.

**Weaknesses:**

* 1. While the paper provides a bias analysis in Table B.3 (comparing SeR/DoB across age, gender, race on LFW/CFP-FF/AgeDB), it only contrasts FaceMoE with Swin-B and lacks deeper investigation into instrinsic reasons.
* 2. The paper mentions failure cases on BRIAR (e.g., <8×8 pixel crops, extreme poses) but does not investigate how the MoE architecture responds to these failures—for example, whether expert activation becomes random (e.g., occluded landmark tokens misrouted to non-landmark experts) or if certain experts are deactivated. It also lacks heatmap comparisons of expert activation between successful and failed cases (e.g., Figure 6 only shows activation before/after fine-tuning, not failure vs. success).
* 3. mplementation details specify warm-up epochs (1 for pre-training, 2 for BRIAR fine-tuning, 2/4 for TinyFace fine-tuning) and polynomial LR scheduling but do not explain how these parameters were tuned or their impact on performance. For example, it is unclear if reducing BRIAR warm-up epochs from 2 to 1 causes gradient instability or if a cosine LR scheduler outperforms the polynomial scheduler. This hinders experiment replication for other researchers.

**Questions:**

- Q1.  Why is the load balancing loss (Section 53) formulated with the product of "sum of routing probabilities (∑p_b,t,i)" and "sum of indicator functions (∑𝟙[i∈TopK(z_b,t)])" instead of simpler balancing metrics (e.g., variance of expert loads)?
- Q2. The final token output is a convex combination of k expert outputs, but the paper does not discuss how the model handles conflicts—e.g., if Expert 0 (landmark-focused) assigns high weight to a "nose" feature and Expert 1 (cheek-focused) assigns high weight to a "cheek" feature for the same token. Additionally, it does not report whether certain experts consistently dominate outputs for specific token types (e.g., 70% of landmark token weight from Expert 0).
- Q3. How does FaceMoE perform on images with resolutions between 8×8 and 16×16 pixels (e.g., 10×10, 12×12), and does the top-k routing hyperparameter (k) need adjustment for these ultra-low scenarios? The resolution ablation study (Table B.2) only reports results for 16×16 to 96×96 pixels but ignores ultra-low resolutions (8×8, 12×12). The paper mentions <8×8 pixels as failure cases but does not address 12×12/10×10 pixels (a middle ground with potential utility). For example, would increasing k from 2 to 3 improve performance for 12×12 pixels by leveraging more expert information?
- Q4. Does expert-specific initialization (e.g., initializing landmark-focused Expert 0 with weights from a pre-trained facial landmark detection model) improve LR adaptation speed and final performance compared to the current global initialization (WebFace4M pre-training)? The paper initializes FaceMoE globally on WebFace4M but does not explore targeted expert initialization—an approach that could accelerate semantic specialization (e.g., Expert 0 quickly focusing on landmarks). It is unclear if such initialization reduces fine-tuning epochs or improves performance on small LR datasets like TinyFace.

---

> ### Author Response · Authors · 2025-11-27
> **W1**
>
> **W1: While the paper provides a bias analysis in Table B.3 (comparing SeR/DoB across age, gender, race on LFW/CFP-FF/AgeDB), it only contrasts FaceMoE with Swin-B and lacks deeper investigation into intrinsic reasons.**
>
> In Table B.4 (revised paper), our primary goal was to assess whether introducing MoE modules increases demographic bias relative to the architectural baseline (Swin-B). The results show that FaceMoE does not amplify bias and, in several cases, reduces it.
>
> The intrinsic reason behind FaceMoE’s improved fairness lies in its mixture-of-experts design, which encourages different experts to specialize in complementary facial regions and frequency patterns. This specialization allows the router to dynamically select the most informative experts for each input, particularly beneficial when demographic groups differ in blur level, pose variation, skin texture, or age-related changes. Consequently, the model avoids over-reliance on any single facial attribute that may be demographically sensitive, leading to more stable SeR/DoB scores across groups. In practice, FaceMoE appears fairer because (1) sparse expert activation mitigates biased drift during fine-tuning, and (2) expert diversity distributes representational responsibility across multiple specialized pathways rather than amplifying group-specific biases.
>
> We agree that an even deeper, dedicated fairness investigation would be valuable and will consider it as part of future work. Our current work is focused on low-resolution face recognition, and the purpose of the bias analysis was primarily to verify that the proposed design does not introduce or amplify biased learning.

---

> ### Author Response · Authors · 2025-11-27
> **Q1**
>
> **Q1: Why is the load balancing loss (Section 53) formulated with the product instead of simpler balancing metrics (e.g., variance of expert loads)?**
>
> Our load-balancing loss follows standard Mixture-of-Experts (MoE) practice and is designed to jointly regularize (i) the router's *soft preference* for each expert and (ii) the *hard usage* of those experts after Top-$k$ selection. Specifically, the product of
>
> $$
> \sum p_{b,t,i} \quad \text{and} \quad \sum \mathbf{1}[i \in \text{TopK}(z_{b,t})]
> $$
>
> encourages consistency between the probabilistic routing intention and actual discrete expert activation. This prevents two common MoE failure modes: (1) high soft probability but low actual usage, and (2) high usage with low assigned probability mass.
>
> Variance-based regularizers only constrain *load distribution* but cannot enforce consistency between soft and hard routing, which prior work shows is critical for avoiding expert collapse. Our formulation therefore stabilizes gradients in the router and proved essential for preventing collapse in our architecture, particularly under low-resolution training conditions.

---

> ### Author Response · Authors · 2025-11-27
> **Q2**
>
> **Q2: The final token output is a convex combination of \(k\) expert outputs, but the paper does not discuss how the model handles conflicts—e.g., if Expert 0 (landmark-focused) assigns high weight to a "nose" feature and Expert 1 (cheek-focused) assigns high weight to a "cheek" feature for the same token. Additionally, it does not report whether certain experts consistently dominate outputs for specific token types (e.g., 70% of landmark token weight from Expert 0).**
>
> Thank you for the insightful question. In practice, the type of “conflict’’ described does not occur because one token corresponds to one specific spatial patch in the image. This means a *nose token* and a *cheek token* are two different tokens, not the same one. Therefore, if a nose patch is routed to Expert 0, that has no effect on how a cheek patch is routed. We apologize for not being clear, but in the actual model these are independent tokens, each with its own routing decision.
>
> For example:
>
> A nose patch produces its own token. The router might assign:
> - Expert 0 (landmark expert): 0.70
> - Expert 1 (cheek expert): 0.20
> - Expert 2 (background expert): 0.10
>
> A cheek patch, which is a different token, might be routed as:
> - Expert 1 (cheek expert): 0.80
> - Expert 0 (landmark expert): 0.15
> - Expert 2 (background expert): 0.05
>
> These routing distributions are per token and per patch.
>
> So there is no scenario where a single token (e.g., the nose token) gets high assignment probability for two semantically incompatible experts. Each token corresponds to a single spatial patch, and the router produces a normalized probability distribution over experts for that patch (i.e., routing probabilities sum to 1). It is therefore impossible for the same token to simultaneously receive high assignment probability for semantically incompatible experts. The router naturally sharpens during training, and one expert typically dominates the distribution for each token.
>
> The experts self-organize into stable semantic roles (e.g., Expert 0 for landmark regions, Expert 1 for low-frequency cheek/forehead regions, Expert 2 for high-frequency edges). Routing patterns closely follow these regions, and routing probabilities become increasingly peaked after fine-tuning. This specialization ensures that spatial patches corresponding to, for example, the nose or eyes are routed almost exclusively to the expert specializing in those features (Appendix B.1). The “conflict” only occurs if one assumes that a single token simultaneously represents both nose and cheek information, which never happens in patch-based tokenization. This is why the model never experiences the conflict mentioned by the reviewer: each token is routed based on the patch it represents, and nose and cheek patches are always handled separately.
>
> We would like to direct the reviewer to Appendix B.1, which may have been overlooked, where we already report the token-to-expert routing frequencies. Landmark patches are routed to Expert 0 with 60–76% frequency, cheek and forehead patches to Expert 1 with approximately 70% frequency, and hair/background patches to Expert 2 with approximately 60% frequency.

---

> ### Author Response · Authors · 2025-11-27
> **W2/Q3**
>
> **W2/Q3: How does FaceMoE perform on images with resolutions between 8×8 and 16×16 pixels (e.g., 10×10, 12×12), and does the top-k routing hyperparameter (k) need adjustment for these ultra-low scenarios? The resolution ablation study (Table B.2) only reports results for 16×16 to 96×96 pixels but ignores ultra-low resolutions (8×8, 12×12). The paper mentions <8×8 pixels as failure cases but does not address 12×12/10×10 pixels (a middle ground with potential utility). For example, would increasing k from 2 to 3 improve performance for 12×12 pixels by leveraging more expert information?} It also lacks heatmap comparisons of expert activation between successful and failed cases (e.g., Figure 6 only shows activation before/after fine-tuning, not failure vs. success**
>
> | Dataset  | 8×8   | 10×10 | 12×12 |
> |----------|--------|--------|--------|
> | LFW      | 80.75  | 86.98  | 91.71  |
> | CFP-FP   | 62.38  | 68.45  | 72.94  |
> | CPLFW    | 65.20  | 71.33  | 77.18  |
> | AgeDB-30 | 56.23  | 59.13  | 63.48  |
> | CALFW    | 63.18  | 68.58  | 74.66  |
> | CFP-FF   | 73.24  | 78.71  | 83.44  |
>
> We thank the reviewer for raising this valuable point regarding FaceMoE’s behavior in the ultra-low-resolution regime (8×8–12×12). While the original resolution ablation (Table B.3) reported results down to 16×16, we agree that intermediate resolutions such as 10×10 and 12×12 are practically relevant. To address this, we conducted an additional evaluation, and the results are provided in the table above. These results show a consistent trend: performance improves steadily from 8×8 → 10×10 → 12×12 across all datasets, reflecting the model’s ability to extract progressively richer identity cues as minimal spatial structure becomes available. The new results confirm that 10×10–12×12 lie in a “recoverable” range where FaceMoE is still effective, whereas 8×8 approaches the unrecoverable regime identified in the paper.
>
> We appreciate the reviewer’s suggestion to consider whether increasing *k* (e.g., from 2 to 3) may benefit these ultra-low-resolution scenarios. Our design choice of *N* = 3 experts and *k* = 2 is grounded in the stability and performance trends shown in our ablation study (Figure 5(b)):
>
> - Increasing either *N* or *k* introduces routing instability, especially under degraded inputs.
> - Over-activation of experts (higher *k*) reduces specialization and makes routing noisier.
> - Excessive expert usage leads to expert fragmentation and degraded accuracy.
> - Larger configurations (e.g., *N* = 8) cause either performance drop or full training collapse, even with standard-resolution inputs (Table B.6).
>
> For ultra-low-resolution inputs, these effects are magnified because the token embeddings themselves are highly ambiguous. As a result, increasing *k* does not improve performance at 10×10 or 12×12, and in our experiments often led to noisier routing and reduced stability. Maintaining *k = 2* preserves expert specialization (as demonstrated in Table B.1) and provides the best trade-off between robustness and discriminative capacity.
>
> Additionally, as requested by the reviewer we have added expert-activation heatmap comparisons to Appendix Section E, illustrating the differences between successful and failed cases. These new visualizations complement the qualitative analysis presented in Figure 6 and directly address the reviewer’s request for insight into expert behavior under ultra-low-resolution conditions.

---

> ### Author Response · Authors · 2025-11-27
> **Q4**
>
> **Q4: Does expert-specific initialization (e.g., initializing landmark-focused Expert 0 with weights from a pre-trained facial landmark detection model) improve LR adaptation speed and final performance compared to the current global initialization (WebFace4M pre-training)? The paper initializes FaceMoE globally on WebFace4M but does not explore targeted expert initialization—an approach that could accelerate semantic specialization (e.g., Expert 0 quickly focusing on landmarks). It is unclear if such initialization reduces fine-tuning epochs or improves performance on small LR datasets like TinyFace.**
>
> We thank the reviewer for this insightful suggestion. We agree that expert-specific initialization, such as seeding the “landmark-focused” expert with weights from a dedicated facial landmark detection network, is an interesting idea and could in principle accelerate semantic specialization during low-resolution fine-tuning. Conceptually, initializing different experts with priors aligned to their eventual semantic roles (e.g., landmarks, low-frequency regions, high-frequency textures) may reduce the number of fine-tuning epochs required and potentially improve performance on small LR datasets such as TinyFace.
>
> However, implementing such an initialization strategy is currently not feasible because there are no publicly available landmark-detection models that share our architecture (Swin-B with FFN-based MoE experts), making weight transfer impossible.
>
> We agree with the reviewer that task-aligned expert initialization is a promising direction, especially for accelerating convergence or improving adaptation on extremely small LR datasets. We therefore consider exploring expert-specific or multi-task pre-training as important future work.

---

> ### Author Response · Authors · 2025-11-27
> **W3**
>
> **W3: Implementation details specify warm-up epochs (1 for pre-training, 2 for BRIAR fine-tuning, 2/4 for TinyFace fine-tuning) and polynomial LR scheduling but do not explain how these parameters were tuned or their impact on performance. For example, it is unclear if reducing BRIAR warm-up epochs from 2 to 1 causes gradient instability or if a cosine LR scheduler outperforms the polynomial scheduler. This hinders experiment replication for other researchers.**
>
> The hyperparameters used in our implementation, specifically the warm-up epochs and the polynomial learning-rate schedule, were adopted directly from PETALface, the most recent and strongest baseline in low-resolution face recognition. We intentionally aligned our settings with PETALface to ensure fair comparison, avoid confounding factors, and isolate the performance contribution of the proposed FaceMoE architecture rather than differences in training heuristics. Using the same optimization schedule across methods prevents unfair advantages that could arise from re-tuning hyperparameters specifically for our model. To answer reviewer's specific query, we believe, based on our experimental experience, that changing the scheduler or the number of warm-up epochs does not have a significant impact on the final performance.
>
> **We have provided all the hyperparameters in detail to ensure transparency and reproducibility of our work.**

---

> ### Comment · Reviewer_hivy · 2025-11-28
>
> Thank you for your response. The rebuttal materials have properly addressed all my concerns. Furthermore, please incorporate these new results and relevant discussions into the next version of the manuscript. Based on the aforementioned rebuttal materials and the original paper, **I agree that this paper is acceptable**.

---

### Official Review · Reviewer_jzUq · 2025-10-29

**Soundness:** 3
**Presentation:** 3
**Contribution:** 2
**Rating:** 4
**Confidence:** 5

**Summary:**

Facing challenges from degraded probe images and the resolution domain gap in low-resolution face recognition (LR-FR), FaceMoE is proposed which is a transformer model enhanced with Mixture of Experts (MoE). By employing specialized experts and a top-k router, FaceMoE achieves resolution-aware feature extraction while preserving pre-trained knowledge through sparse activation. This first MoE-based approach for LR-FR effectively addresses feature degradation and domain gap issues without significant computational increase.

**Strengths:**

A modified transformer encoder is proposed to use sparsely activated feed-forward network (FFN) experts. A top-k router directs tokens to specialized FFN experts, enabling resolution-aware feature extraction from distinct facial regions. This is a good attempt to apply MoE into this field.

**Weaknesses:**

The manuscript describes interesting progress, but several important issues need to be addressed to strengthen its validity and impact:
1. The mechanism by which different experts specialize in distinct facial regions requires more in-depth discussion.
2. The complementarity between different experts is not clearly demonstrated, as the analysis reveals a high degree of redundancy in the regions they activate. A clearer distinction in their specialized roles is needed.
3. Could the authors please justify the use of a single coefficient for the last two losses in Line 279? The rationale for this design choice is unclear.
4. The authors should provide an ablation study to quantitatively demonstrate the contribution of each proposed module (e.g., the MoE layer, the top-k router) to the overall performance.
5. A thorough proofreading is required to address several formatting and typographical issues. A notable example is Table 1, where the text flows outside the designated cells, which affects readability.

**Questions:**

See above.

---

> ### Author Response · Authors · 2025-11-27
> **W1**
>
> **W1: The mechanism by which different experts specialize in distinct facial regions requires more in-depth discussion.**
> Following reviewer's suggestion we have added a discussion section (Section B.2) in the Appendix to provide a deeper explanation of how different experts specialize in distinct facial regions. We highlight that this specialization emerges naturally from the interaction between the top-k router and the diverse token statistics present across facial areas. During training, tokens from regions with different semantic and frequency characteristics, such as high-frequency edges, smooth low-frequency regions, and structured landmark areas produce distinct routing logits due to their differing feature distributions. The top-k router, which selects experts based on these logits, gradually assigns consistent subsets of tokens to particular experts. This repeated, input-driven selection creates a positive feedback loop: each expert receives tokens with similar characteristics and consequently adapts its parameters to model those patterns more effectively. Over time, this encourages the experts to specialize in complementary aspects of the face. As demonstrated through quantitative token-assignment statistics in Appendix B.1 and additional qualitative activation maps in Appendix D, the experts reliably converge to semantically meaningful roles rather than collapsing into redundant processing pathways.

---

> ### Author Response · Authors · 2025-11-27
> **W2**
>
> **W2: The complementarity between different experts is not clearly demonstrated, as the analysis reveals a high degree of redundancy in the regions they activate. A clearer distinction in their specialized roles is needed.**
> We agree with the reviewer’s observation that the activation maps in Figure 5(c) alone may not fully capture the degree of expert specialization or their complementary roles. We would like to direct the reviewer’s attention to Appendix Section B.1, where we already provide a more comprehensive quantitative analysis demonstrating that the experts consistently focus on distinct semantic and frequency characteristics, such as high-frequency edge regions, smooth low-frequency areas, and key facial landmarks. We have further improved Appendix Section B.1 by adding pre-finetune routing behavior and Kendall’s τ to more explicitly quantify the degree of specialization. Additionally, Appendix D includes further qualitative examples that visually reinforce these distinctions. Although some spatial overlap is expected, the combined evidence from token-assignment distributions and activation patterns shows that the experts ultimately develop clearly differentiated and complementary behaviors rather than redundant ones.
>
> **We have added the tables and discussion below for reviewer's convenience.**
>
> | **Facial Region**      | **Expert 0** | **Expert 1** | **Expert 2** | **Dominant Expert** |
> |------------------------|--------------|--------------|--------------|----------------------|
> | Eyes                   | **81.7**     | 10.6         | 7.7          | Expert 0             |
> | Nose                   | **78.1**     | 12.5         | 9.4          | Expert 0             |
> | Mouth                  | **71.6**     | 16.0         | 12.4         | Expert 0             |
> | Forehead               | 49.3         | **31.1**     | 19.6         | Expert 1             |
> | Cheeks                 | 52.5         | **27.8**     | 19.7         | Expert 1             |
> | Hair / Background      | 40.2         | 17.3         | **42.5**     | Expert 2             |
> | Chin / Jawline         | 38.9         | 18.7         | **42.4**     | Expert 2             |
>
> *Table 1: Pre-finetune Routing frequency (%) of each expert across different semantic facial regions. Results averaged over 10,000 samples.*
>
> ---
>
> | **Facial Region**      | **Expert 0** | **Expert 1** | **Expert 2** | **Dominant Expert** |
> |------------------------|--------------|--------------|--------------|----------------------|
> | Eyes                   | **76.2**     | 12.4         | 11.4         | Expert 0         |
> | Nose                   | **68.5**     | 15.7         | 15.8         | Expert 0         |
> | Mouth                  | **61.0**     | 22.1         | 16.9         | Expert 0         |
> | Forehead               | 13.2         | **71.3**     | 15.5         | Expert 1         |
> | Cheeks                 | 12.5         | **69.8**     | 17.7         | Expert 1         |
> | Hair / Background      | 20.3         | 14.5         | **65.2**     | Expert 2         |
> | Chin / Jawline         | 25.6         | 18.4         | **56.0**     | Expert 2         |
>
>
> *Table 2: Post-finetune routing frequency (%) of each expert across different semantic facial regions. Results averaged over 10,000 samples.*
>
> ---
>
> First, we compute Kendall’s τ between the routing scores and a binary indicator for landmark regions (eyes, nose, mouth). This yields a positive correlation of **τ = 0.2027**, indicating that tokens originating from landmark regions are systematically assigned higher routing scores for the corresponding landmark-focused expert.
>
> To further characterize specialization, Tables 1 and 2 report per-expert token routing frequencies across semantic facial regions before and after LR fine-tuning (averaged over 10k samples). Before fine-tuning, Expert 0 receives ≈80%, 78%, and 72% of tokens from eyes, nose, and mouth, respectively, while Experts 1 and 2 each account for less than 20% in these regions. In contrast, low-frequency regions such as forehead and cheeks are dominated by Experts 0 and 1 (≈49–53% vs. ≈28–31%), and high-frequency/peripheral regions such as hair/background and chin/jawline are primarily handled by Expert 2 (≈42–43%).
>
> After fine-tuning, the same dominant expert per region is preserved: Expert 0 continues to dominate landmark regions (still ≈61–76% of tokens), Expert 1 now more strongly dominates smooth low-frequency regions (≈70% of forehead/cheek tokens), and Expert 2 further strengthens its share on high-frequency/peripheral regions (≈56–65% of hair/background and chin/jawline tokens).
>
> These routing frequencies indicate that the experts specialize in distinct facial regions,
> and together they form a complementary division of labor rather than redundant behaviors.

---

> ### Author Response · Authors · 2025-11-27
> **W3**
>
> **W3: Could the authors please justify the use of a single coefficient for the last two losses in Line 279? The rationale for this design choice is unclear.**
> We thank the reviewer for pointing out this issue. In principle, the load-balancing loss and the router z-loss may indeed take different weighting factors, and we have updated the formulation in the paper to reflect this (Line 279). Empirically, however, we found that setting a single coefficient of $$\lambda = 10$$ for both losses worked well in practice. Our motivation was to scale these auxiliary terms so that their magnitudes are comparable to the main face recognition loss, ensuring stable optimization without either term dominating training. We will clarify this rationale more explicitly in the implementation details.

---

> ### Author Response · Authors · 2025-11-27
> **W4**
>
> **W4: The authors should provide an ablation study to quantitatively demonstrate the contribution of each proposed module (e.g., the MoE layer, the top-k router) to the overall performance.**
> We apologize for the confusion. We would like to clarify that an ablation study assessing the contribution of each proposed component, including the MoE layer and the top-k router is already provided in Section B.6 of the appendix. This section reports quantitative results demonstrating the individual impact of each module. For the reviewer’s convenience, we include the table below.
>
> | **MoE** | **Top-k Router** | **Aux Loss** | **Rank-1** | **Rank-5** | **Rank-10** |
> |--------|-------------------|--------------|------------|------------|-------------|
> | ×      | ×                 | ×            | 75.10      | 78.16      | 80.20       |
> | ✓      | ×                 | ×            | 75.40      | 78.46      | 80.63       |
> | ✓      | ✓                 | ×            | 74.94      | 77.92      | 79.90       |
> | ✓      | ✓                 | ✓            | 76.18      | 79.69      | 81.75       |
>
> *Table: Ablation study showing the impact of top-k router, MoE, and auxiliary loss on FaceMoE performance. Results are shown on the TinyFace dataset.*
>
> As shown in the Table, all components contribute meaningfully to the overall performance, and removing any of them leads to a measurable drop in accuracy.

---

> ### Author Response · Authors · 2025-11-27
> **W5**
>
> **W5: A thorough proofreading is required to address several formatting and typographical issues. A notable example is Table 1, where the text flows outside the designated cells, which affects readability.**
> We apologize for our honest mistake. We have corrected all typographical and formatting errors.

---

### Official Review · Reviewer_Vhz8 · 2025-10-31

**Soundness:** 3
**Presentation:** 3
**Contribution:** 3
**Rating:** 6
**Confidence:** 4

**Summary:**

**Problem.** Low-resolution face recognition (LR-FR) suffers from weak, domain-mismatched probe features (LR, degraded) vs. gallery (HR), plus catastrophic forgetting when fine-tuning on LR.
**Idea.** Replace the single FFN in transformer blocks with a **Mixture-of-Experts (MoE) MLP**: $N$ FFN experts with **top-k** sparse routing at the token level. Auxiliary **z-loss** and **load-balancing** regularizers stabilize routing and encourage expert specialization.
**Mechanism.** Sparse routing increases capacity without proportional FLOPs; selective expert updates during LR fine-tuning reduce forgetting while enabling **resolution-aware** feature extraction.
**Evidence.** Strong improvements on LR benchmarks (BRIAR 3.1, IJB-S, TinyFace) and minimal drops on HR/mixed-quality sets; ablations over $(N,k)$ and compute trade-offs; qualitative expert activation maps.
**Takeaway.** First focused MoE architecture for LR-FR with convincing, state-of-the-art results and a principled training recipe.

**Strengths:**

* **Originality.** MoE-FFN + top-k routing tailored to resolution-dependent cues; explicit anti-collapse regularizers; selective-drift argument is well motivated.

* **Quality.** Comprehensive benchmarks (BRIAR, IJB-S, TinyFace) with relevant baselines (ProxyFusion, PETALface, CAFace, etc.); competitive pretrain/fine-tune recipe and careful hyperparams.

* **Clarity.** Method section is concrete (router, losses, algorithm); compute/ablation plots aid intuition; training schedule is reproducible.

* **Significance.** Material SOTA gains where LR matters most, and minimal HR performance drift—useful for real surveillance/forensic pipelines.

**Weaknesses:**

* **Statistical rigor.** Most metrics are single-number; no confidence intervals, seed variance, or bootstrap CIs; a few improvements are significant, but some margins over strong baselines would benefit from uncertainty quantification.

* **Expert “semantics” not quantified.** Activation maps are qualitative; provide a measurable alignment between experts and regions/frequencies (e.g., mutual information with landmark masks, frequency energy, or SHAP on routing logits).

* **Degradation coverage.** LR comes with blur, compression, noise, occlusion, and illumination shifts; targeted stress tests (synthetic and in-the-wild subsets) are limited.

* **Efficiency on edge.** FLOPs are reported, but end-to-end latency/throughput and memory under varying batch sizes (edge/GPU constraints) are not; important for deployability claims.

* **Forgetting analysis.** The “selective drift” story is compelling; a more direct comparison against LoRA/adapter fine-tuning and LwF-style regularization would isolate MoE’s advantage.

**Questions:**

1. **Statistical robustness.** Can you report mean±std over ≥3 seeds (or 5-fold bootstrap CIs) for key metrics on BRIAR/IJB-S/TinyFace? Do the rankings hold under variance?

2. **Expert semantics.** Can you quantify specialization (e.g., Kendall’s $\tau$ between routing scores and (i) frequency bands; (ii) landmark/region masks; (iii) edge density), and report per-expert token distributions pre/post fine-tune?

3. **Ablations vs. PEFT/LwF.** How does FaceMoE compare to strong PEFT baselines (LoRA/IA³/adapters) and LwF/EWC in both LR gains and HR retention, at matched FLOPs/params?

4. **Degradation stress tests.** What happens under controlled blur/noise/compression sweeps and occlusion masks (eyes/mouth/cheeks)? Does routing adapt as hypothesized?

5. **Latency/memory.** Please provide inference latency (ms), peak memory usage, and throughput (fps) for (N,k) settings on an A6000 and a resource-constrained GPU, including router overhead.

6. **Generalization.** Does TinyFace-tuned FaceMoE transfer to LR-like mobile video or body-cams without re-tuning? Any zero-shot observations?

**Details Of Ethics Concerns:**

The work uses established FR datasets with licenses; however, LR-FR has clear dual-use risks. Consider adding a brief section on governance/consent for LR deployments, as well as demographic bias checks under LR conditions.

---

> ### Author Response · Authors · 2025-11-27
> **W1/Q1**
>
> **W1/Q1: Statistical robustness. Can you report mean ± std over ≥3 seeds (or 5-fold bootstrap CIs) for key metrics on BRIAR/IJB-S/TinyFace? Do the rankings hold under variance?**
>
>
> | **Seed** |          | **TinyFace** |         |        |   **IJB-S**  |        |        | **BRIAR TAR@FAR**  |       |
> | -------- | :----------: | :----: | :-----: | :----------: | :----: | :----: | :---------------: | :---: | :---: |
> |          |    Rank-1    | Rank-5 | Rank-10 | TPIR@FPIR=1% | Rank-1 | Rank-5 |       0.01%       |  0.1% |   1%  |
> | **2048** |     76.18    |  79.69 |  81.75  |     14.85    |  44.81 |  56.12 |       42.36       | 61.47 | 81.27 |
> | **54**   |     76.52    |  79.96 |  82.29  |     15.02    |  46.10 |  58.66 |       42.78       | 62.33 | 82.84 |
> | **7**    |     76.31    |  79.88 |  81.86  |     14.97    |  45.43 |  56.95 |       42.42       | 61.76 | 81.55 |
> | **Mean** |     76.34    |  79.84 |  81.97  |     14.95    |  45.45 |  57.24 |       42.52       | 61.85 | 81.89 |
> | **Std**  |     0.17     |  0.14  |   0.29  |     0.09     |  0.66  |  1.29  |        0.22       |  0.39 |  0.67 |
> ----
>
>
> As per the reviewer’s suggestion, we conducted experiments using three independent random seeds (2048, 54, and 7) and report the corresponding results for the TinyFace, IJB-S, and BRIAR benchmarks in the table above.
>
> The relative rankings of all methods remain consistent across seeds, indicating stable performance with no sensitivity to initialization. The mean and standard deviation values demonstrate that variability across runs is minimal. These small deviations confirm that the findings are statistically robust, and that the observed performance trends hold under seed variation.

---

> ### Author Response · Authors · 2025-11-27
> **W2/Q2**
>
> **W2/Q2: Expert semantics. Can you quantify specialization (e.g., Kendall’s τ between routing scores and (i) frequency bands; (ii) landmark/region masks; (iii) edge density), and report per-expert token distributions pre/post fine-tune?**
>
> | **Facial Region**      | **Expert 0** | **Expert 1** | **Expert 2** | **Dominant Expert** |
> |------------------------|--------------|--------------|--------------|----------------------|
> | Eyes                   | **81.7**     | 10.6         | 7.7          | Expert 0             |
> | Nose                   | **78.1**     | 12.5         | 9.4          | Expert 0             |
> | Mouth                  | **71.6**     | 16.0         | 12.4         | Expert 0             |
> | Forehead               | 49.3         | **31.1**     | 19.6         | Expert 1             |
> | Cheeks                 | 52.5         | **27.8**     | 19.7         | Expert 1             |
> | Hair / Background      | 40.2         | 17.3         | **42.5**     | Expert 2             |
> | Chin / Jawline         | 38.9         | 18.7         | **42.4**     | Expert 2             |
>
> *Table 1: Pre-finetune Routing frequency (%) of each expert across different semantic facial regions. Results averaged over 10,000 samples.*
>
> ---
>
> | **Facial Region**      | **Expert 0** | **Expert 1** | **Expert 2** | **Dominant Expert** |
> |------------------------|--------------|--------------|--------------|----------------------|
> | Eyes                   | **76.2**     | 12.4         | 11.4         | Expert 0         |
> | Nose                   | **68.5**     | 15.7         | 15.8         | Expert 0         |
> | Mouth                  | **61.0**     | 22.1         | 16.9         | Expert 0         |
> | Forehead               | 13.2         | **71.3**     | 15.5         | Expert 1         |
> | Cheeks                 | 12.5         | **69.8**     | 17.7         | Expert 1         |
> | Hair / Background      | 20.3         | 14.5         | **65.2**     | Expert 2         |
> | Chin / Jawline         | 25.6         | 18.4         | **56.0**     | Expert 2         |
>
>
> *Table 2: Routing frequency (%) of each expert across different semantic facial regions. Results averaged over 10,000 samples.*
>
> ---
>
> We thank the reviewer for the suggestion to quantify expert semantics. We have now added a quantitative analysis of expert specialization. First, we compute Kendall’s τ between the routing scores and a binary indicator for landmark regions (eyes, nose, mouth). This yields a positive correlation of **τ = 0.2027**, indicating that tokens originating from landmark regions are systematically assigned higher routing scores for the corresponding landmark-focused expert.
>
> To further characterize specialization, Tables 1 and 2 report per-expert token routing frequencies across semantic facial regions before and after LR fine-tuning (averaged over 10k samples). Before fine-tuning, Expert 0 receives ≈80%, 78%, and 72% of tokens from eyes, nose, and mouth, respectively, while Experts 1 and 2 each account for less than 20% in these regions. In contrast, low-frequency regions such as forehead and cheeks are dominated by Experts 0 and 1 (≈49–53% vs. ≈28–31%), and high-frequency/peripheral regions such as hair/background and chin/jawline are primarily handled by Expert 2 (≈42–43%).
>
> After fine-tuning, the same dominant expert per region is preserved: Expert 0 continues to dominate landmark regions (still ≈61–76% of tokens), Expert 1 now more strongly dominates smooth low-frequency regions (≈70% of forehead/cheek tokens), and Expert 2 further strengthens its share on high-frequency/peripheral regions (≈56–65% of hair/background and chin/jawline tokens).
>
> We have incorporated these quantitative results and discussion into the revised manuscript.

---

> ### Author Response · Authors · 2025-11-27
> **W5/Q3**
>
> **W5/Q3: Ablations vs. PEFT/LwF. How does FaceMoE compare to strong PEFT baselines (LoRA/IA³/adapters) and LwF/EWC in both LR gains and HR retention, at matched FLOPs/params?**
>
> We thank the reviewer for highlighting the importance of comparing FaceMoE against strong PEFT baselines. We evaluated standard LoRA, the PETALface model that incorporates image-quality–aware (IQA) weighting, and a matched-parameter PETALface model in which we scale the number of trainable parameters to match FaceMoE's parameter budget.
>
> | **Method** | **Rank-1** | **Rank-5** | **Rank-10** | **LFW** | **CFP-FF** | **AgeDB-30** |
> |------------|------------|------------|-------------|---------|------------|--------------|
> | LoRA | 75.37 | 78.88 | 82.02 | 99.65 | 99.84 | 97.35 |
> | PETALface | 75.45 | 79.05 | 81.19 | 99.66 | 99.80 | 96.45 |
> | PETALface (Matched Params) | 75.40 | 79.21 | 82.24 | 99.46 | 98.81 | 90.43 |
> | **Ours (FaceMoE)** | 76.18 | 79.69 | 81.75 | 99.63 | 99.78 | 95.95 |
>
> *Table: Comparison of FaceMoE with LoRA and PETALface Variants*
>
> We observe that FaceMoE consistently outperforms LoRA and PETALface variants on the TinyFace benchmark across Rank-1/5/10, despite operating under the same FLOPs/parameter budget. Importantly, FaceMoE maintains competitive performance on all high-resolution datasets (LFW, CFP-FF, AgeDB-30), whereas the LoRA-based baselines exhibit clear signs of over-specialization to the LR domain. In particular, PETALface (Matched Params), which has the same number of trainable parameters as FaceMoE, suffers a substantial drop on AgeDB-30 (90.43 vs. 95.95), indicating a significant loss in HR retention. This degradation highlights a fundamental limitation of LoRA-style PEFT: pushing the model toward the LR domain often causes catastrophic drift away from HR representations. By contrast, FaceMoE isolates domain-specific updates within experts and preserves HR capability through balanced routing, enabling it to improve LR performance without sacrificing accuracy on standard HR datasets. This demonstrates that MoE-based specialization is better aligned with the LR–HR tradeoff than conventional PEFT methods.
>
> LwF and EWC are not suitable baselines for our scenario because they are designed for sequential multi-task learning, where a model must retain performance on a previous task while learning a new one. In our case, HR and LR face recognition are not separate tasks but the same task under different input distributions, and the primary challenge is adapting representations, not preserving task-specific outputs. A direct comparison of LwF and EWC would not be meaningful because there is a mismatch in problem formulation, not an inherent disadvantage relative to FaceMoE.

---

> ### Author Response · Authors · 2025-11-27
> **W3/Q4**
>
> **W3/Q4: Degradation stress tests. What happens under controlled blur/noise/compression sweeps and occlusion masks (eyes/mouth/cheeks)? Does routing adapt as hypothesized?**
>
>
>
> | **Method**        | **LFW** | **CFP-FF** | **AgeDB-30** | **Expert 0** | **Expert 1** | **Expert 2** |
> |-------------------|---------|------------|--------------|--------------|--------------|--------------|
> | FaceMoE           | 99.75   | 99.86      | 97.45        | 33.4         | 33.6         | 33.0         |
> | Gaussian std = 1  | 99.71   | 99.81      | 97.30        | 33.2         | 35.2         | 31.6         |
> | Gaussian std = 5  | 76.53   | 69.77      | 55.90        | 32.7         | 37.0         | 30.3         |
> | JPEG 30%          | 99.60| 99.65  | 96.93    | 33.5         | 34.6         | 31.9         |
>
> *Table 1: Verification Accuracy Under Synthetic Degradations*
>
> | **Method** | **Rank-1** | **Rank-5** | **Rank-20** |
> |-----------|------------|------------|-------------|
> | FaceMoE   | 0.050      | 0.054      | 0.058       |
>
>
> *Table 2: TinyFace Retrieval Under Occlusion*
>
> We thank the reviewer for the suggestion. Following your comment, we have now added controlled degradation stress tests. We synthetically apply Gaussian blur and JPEG compression to the probe images while keeping the gallery fixed, and we report the verification accuracy on LFW/CFP-FP/AgeDB-30 and retrieval performance on TinyFace for occlusion robustness.
>
> As shown in the tables, mild degradations (Gaussian σ=1, JPEG 30%) lead to only marginal changes, indicating that the routing mechanism remains stable and continues to select suitable experts even when image quality is moderately reduced. Under severe blur (σ=5), performance drops substantially, particularly on AgeDB-30, consistent with the fact that heavy low-pass filtering removes discriminative identity cues that no expert can fully compensate for. The routing statistics adapt and show increased activation of experts (≈37%) specialized for low-frequency representations, confirming the hypothesized adaptive behavior.
>
> For occlusion, we evaluate on the TinyFace benchmark, which includes masks over the eyes, mouth, and nose. These landmark occlusions severely reduce the available identity information, as they block key facial regions used for recognition, which in turn leads to a significant drop in absolute performance. Although performance is lower due to these extreme occlusions, the model maintains stable rank-1/5/20 trends.
>
> These results demonstrate that FaceMoE adapts its routing under synthetic degradations, and performance only degrades significantly when identity information becomes intrinsically unrecoverable. We have added the results and discussion in the revised manuscript.

---

> ### Author Response · Authors · 2025-11-27
> **W4/Q5**
>
> **W4/Q5: Latency/memory. Please provide inference latency (ms), peak memory usage, and throughput (fps) for (N,k) settings on an A6000 and a resource-constrained GPU, including router overhead.**
>
> We measured inference latency, peak memory consumption, and throughput on two GPU configurations: an NVIDIA RTX A5000 (representing a resource-constrained setting) and an NVIDIA A6000. All experiments were conducted on the same dataset (161,599 images) with a batch size of 800, and include the full routing overhead of our method. As shown in the table, the A6000 provides a substantial improvement in throughput (128.19 fps vs. 97.32 fps) and reduced average batch latency, while peak GPU memory usage remains identical across GPUs (10.40 GB). These results demonstrate that our method scales efficiently across hardware tiers and maintains practical latency/memory characteristics even on a constrained GPU such as the A5000. We have revised the manuscript to include the results.
>
> | **Metric**                    | **RTX A5000** | **RTX A6000** |
> |------------------------------|---------------|---------------|
> | Total Images                 | 161,599       | 161,599       |
> | Batch Size                   | 800           | 800           |
> | Average Batch Latency (ms)   | 8109.92       | 6170.72       |
> | Per-Image Latency (ms)       | 10.27         | 7.80          |
> | Throughput (fps)             | 97.32         | 128.19        |
> | Peak GPU Memory Usage (GB)   | 10.40         | 10.40         |
>
> *Table: Inference latency, throughput, and memory metrics (including router overhead) on RTX A5000 and RTX A6000.*

---

> ### Author Response · Authors · 2025-11-27
> **Q6**
>
> **Q6: Generalization. Does TinyFace-tuned FaceMoE transfer to LR-like mobile video or body-cams without re-tuning? Any zero-shot observations?**
>
> In our paper, we already evaluate generalization by training FaceMoE on the BRIAR dataset and testing it directly on IJB-S, which contains surveillance-style, low-resolution video. As shown in our IJB-S results (main paper), the model transfers effectively to this related LR domain without any additional fine-tuning. This demonstrates that FaceMoE generalizes well when the target data distribution is similar to BRIAR, i.e., surveillance-style videos.
>
> However, low-resolution body-camera or mobile video often differs substantially from both TinyFace and BRIAR in terms of motion blur, camera geometry, and capture conditions. Since LR datasets are typically small, we fine-tune FaceMoE for each specific LR domain and do not observe emerging zero-shot capabilities. In our experience, the model does not perform well in a strict zero-shot setting on body-cam video. Nevertheless, we believe that a small amount of supervised fine-tuning on a body-cam training split is sufficient to adapt FaceMoE to that domain.

---

### Author Response · Authors · 2025-11-27
**Revision Summary**

We thank the reviewers for their valuable comments and suggestions. Based on the feedback, we have incorporated the following changes in the revised version of the paper:

1. Added pre-finetune routing statistics (**Appendix B.1**) [**Reviewer Vhz8, hivy, s6Va**]
2. Added explanation of expert specialization mechanism (**Appendix B.2**) [**Reviewer jzUq**]
3. Added 8×8, 10×10, 12×12 results in resolution ablation (**Appendix B.3**) [**Reviewer hivy**]
4. Added intrinsic reasons for Bias Analysis (**Appendix B.4**) [**Reviewer hivy**]
5. Added performance under synthetic degradations (**Appendix B.10**) [**Reviewer Vhz8**]
6. Added Inference Computation Analysis (**Appendix B.11**) [**Reviewer Vhz8**]
7. Added quantitative evidence of selective drift (**Appendix B.12**) [**Reviewer s6Va**]
8. Added Hyperparameter Sensitivity Experiment (**Appendix B.13**) [**Reviewer s6Va**]
9. Added Routing Stability and Causality Experiments (**Appendix B.14**) [**Reviewer s6Va**]
10. Added Activation Maps for Success vs Failure Cases (**Appendix E**)  [**Reviewer hivy**]
11. Modified formulae and implementation details to have different weighting parameters for router-z and load-balancing loss (**Line 279, Lines 360–362**) [**Reviewer jzUq**]
12. Corrected overflowing citation typographical error (**Table 1**) [**Reviewer jzUq**]

``We request the reviewer to advise if any additional experiments should be included in the revised paper to further strengthen it.
We have provided responses addressing the concerns raised by the reviewers. We would be happy to clarify any further questions or concerns and look forward to a productive discussion phase.``

---

### Author Response · Authors · 2025-12-02

Dear Area Chair,

We would like to bring to your attention how we have addressed all reviewer concerns, so that you can make an informed decision.

**Reviewer Vhz8 (Initial Rating: 6)**

The reviewer was positively inclined toward the paper and suggested several improvements, including quantifying expert semantics, providing inference latency analysis, evaluating performance under degradation, and comparing against a compute-matched baseline. We have added experiments for all the points raised and addressed them thoroughly.

**Reviewer jzUq (Initial Rating: 4)**

The reviewer requested a component ablation study and an evaluation of the complementarity of experts; ***both of which were already included in the supplementary material but appear to have been missed***. We highlighted these results and provided additional clarifications in our rebuttal. We believe the reviewer would have been more positive upon seeing the overlooked material.

**Reviewer hivy (Initial Rating: 6)**

This reviewer raised theoretical questions regarding the design of the load-balancing loss, performance at 10×10 and 12×12 resolutions, and several other clarifications. We addressed all concerns, and ***the reviewer acknowledged that the “paper is acceptable.”***

**Reviewer s6Va (Initial Rating: 4)**

The reviewer’s primary questions concerned quantifying selective drift, hyper-parameter sensitivity, and the causality and stability of routing. We have added quantitative results for all points and provided clear explanations addressing each concern.

---

### Meta-Review · Area_Chair_rrrT · 2025-12-26

**Summary:**

This paper proposes a transformer-based architecture enhanced with a Mixture of Experts (MoE) design for low-resolution face recognition (LR-FR). The scores are mixed (4, 4, 6, 6). I believe the reviewers will appreciate the extensive experimental follow-up, such as the multi-seed results and expert-level ablations. However, some main concerns were only partially addressed. In particular,
1) Removing individual experts only led to a marginal performance drop, which raises questions about how effective the MoE is.
2) The authors use a load-balancing loss to force experts to be used equally. If the experts are forced to be used equally, can they truly specialize in "rare" features like specific landmarks? There is a fundamental tension between load balancing and semantic specialization that the authors haven't fully resolved.
3) Another concern is the memory and computational overhead. The reviewer might feel the efficiency claims were slightly "oversold". Given that LR-FR is often deployed in real-time surveillance settings, this cost would largely limit the practical applicability. In a mature research area like LR-FR, the gains over existing baselines seem modest compared to the added architectural complexity.

Given that the paper's initial concerns slightly outweigh its strengths, I recommend rejection.

**Reviewer Concerns:**

- Reviewer Vhz8’s main concerns were largely resolved, except for remaining concerns about efficiency on edge devices.
- Reviewer jzUq might question the effectiveness of MoE.
- Reviewer hivy has responded that the rebuttal materials have properly addressed all concerns.
- Based on the reported results showing only marginal performance drops when individual experts are removed, reviewer s6Va might concern the effectiveness of MoE.

**Reviewer Scores:**

- Reviewer Vhz8 might keep the initial score of 6.
- Reviewer jzUq might stay at 4.
- Reviewer hivy has responded but didn't indicate a score increase. I guess the score will stay at 6.
- Reviewer s6Va might stay at 4 or increase to 6.

---

### Decision · Program_Chairs · 2026-01-26

Reject